# Study on the Effect of Polycarboxylate Ether Molecular Structure on Slurry Dispersion, Adsorption, and Microstructure

**DOI:** 10.3390/polym15112496

**Published:** 2023-05-29

**Authors:** Yunhui Fang, Zhijun Lin, Dongming Yan, Xiaofang Zhang, Xiuxing Ma, Junying Lai, Yi Liu, Zhanhua Chen, Zhaopeng Wang

**Affiliations:** 1Polytechnic Institute, Zhejiang University, Hangzhou 310015, China; fangyunhui@126.com; 2College of Civil Engineering and Architecture, Zhejiang University, Hangzhou 310058, China; dmyan@zju.edu.cn; 3KZJ New Materials Group Co., Ltd., Xiamen 361101, China; geniusvcap@163.com (Z.L.); 15859120891@163.com (X.Z.); mary@xmabr-kzj.com (X.M.); zhanhuachen123@126.com (Z.C.); wangzp1109@126.com (Z.W.); 4School of Materials Science and Engineering, Zhejiang University, Hangzhou 310023, China; liuyimse@zju.edu.cn

**Keywords:** PCE, GPC, structure, carboxyl density, main chain polymerization degree, adsorption, hydration kinetics

## Abstract

This study synthesized polycarboxylate superplasticizer (PCE) with varying carboxyl densities and main chain degrees of polymerization. The structural parameters of PCE were characterized using gel permeation chromatography and infrared spectroscopy. The study investigated the impact of PCE’s diverse microstructures on cement slurry’s adsorption, rheology, hydration heat, and kinetics. Microscopy was used to analyze the products’ morphology. The findings indicated that an increase in carboxyl density led to an increase in molecular weight and hydrodynamic radius. A carboxyl density of 3.5 resulted in the highest flowability of cement slurry and the most considerable adsorption amount. However, the adsorption effect weakened when the carboxyl density was the highest. Decreasing the main chain degree of polymerization led to a significant reduction in the molecular weight and hydrodynamic radius. A main chain degree of 16.46 resulted in the highest flowability of slurry, and both large and small main chain degrees of polymerization exhibited single-layer adsorption. PCE samples with higher carboxyl density caused the greatest delay in the induction period, whereas PCE-3 promoted the hydration period’s acceleration. Hydration kinetics model analysis indicated that PCE-4 yielded needle-shaped hydration products with a small nucleation number in the crystal nucleation and growth stage, while PCE-7’s nucleation was most influenced by ion concentration. The addition of PCE improved the hydration degree after three days and facilitated the strength’s later development compared to the blank sample.

## 1. Introduction

Polycarboxylate superplasticizer (PCE) is widely used in the concrete industry due to its excellent properties such as high water reduction rate, workability, and low slump loss, essential for creating high fluidity, strength, and self-compacting concrete [1]. PCE has become the fifth component of modern concrete, apart from cement, water, coarse aggregate, and fine aggregate [2]. PCE, which has a comb-like structure, adsorbs on the surface of cement particles through the anionic carboxyl group on the main chain, producing electrostatic repulsion and spatial hindrance, leading to the dispersion of cement [3,4]. At the same time, its side chain stretches between the pore solution and different cement particles, producing a spatial hindrance effect [5], which brings good dispersion of cement, especially in the case of a low water-cement ratio, and has a good retarding effect [1,6,7]. The dispersibility and dispersion retention of PCE in cement is significantly affected by the differences in the composition of different concrete components, such as cement and fine aggregate and changes in environmental temperature [8,9,10]. The high clay content in sand significantly reduces the performance of newly mixed concrete containing PCE [11]. High temperature affects the structure of PCE and may even destroy some of the structures [12].

Many scholars have designed PCE with different molecular structures and have optimized its performance using different functional groups and different structural arrangements to meet various concrete performance requirements [13,14,15,16]. The introduction of different functional groups in superplasticizers results in different adsorption properties. The introduction of sulfonic acid groups can quickly adsorb on cement particles and improve the adsorption behavior of PCE, while the introduction of ester and acyl groups reduces the adsorption performance of PCE [15]. The chemical structure of superplasticizers affects the dispersibility of cement. The grafting chain length is highly dispersed and stable when the adsorption amount is small, and the grafting chain length increases with the increase of the adsorption density, which affects the spatial structure of the superplasticizer [13]. Frank Winnefeld found that reducing the side chain density of superplasticizers can improve the workability of mortar, while side chain length and molecular weight have little effect because PCE chains exhibit a non-stretching, mushroom-like conformation in highly ion-concentrated aqueous solutions [16]. The microstructure of different PCE affects the rheological properties of cement paste, mortar, and concrete. Generally, PCE with lower main chain polymerization degrees and higher sulfonic acid content have lower yield stress and better rheological properties [17]. Long-side chain PCE can effectively reduce the plastic viscosity of mortar, and the molecular weight gradually decreases the residual viscosity of the slurry [18,19].

Different microstructure PCE have different conformations, which lead to different adsorption behaviors on cement particles [18]. The interaction between superplasticizers and cement particles is based on adsorption [20]. The adsorption amount of superplasticizer on cement particle surface is closely related to the molecular structure of superplasticizer. Superplasticizer forms a double electric layer on the surface of cement particles, and their thickness is determined by the density of the superplasticizer adsorbed on the surface of cement particles and the configuration of the superplasticizer. The thickness determines the priority of the steric hindrance effect and electrostatic repulsion to the dispersion effect [21], as shown in Figure 1.

Liu et al. found that two comb-shaped copolymers with the same side chain length and bridging groups of ester and ether exhibited similar adsorption characteristics on silicate surfaces, while the ester group had a higher adsorption rate on the hydration product of calcium sulfoaluminate (AFt) [22]. The adsorption amount of water reducer is easily affected by sulfate and other adsorption groups. Sylvie Pourchet studied the effect of the redistribution of negative charge groups/PEG side chains in polycarboxylate comb-shaped copolymers on the dispersibility of comb-shaped copolymers, indicating that “gradient” copolymers are less sensitive to sulfate competitive adsorption [5]. Different types of water reducers have different adsorption characteristics on the surface of cement particles, and there are multiple adsorption models, such as the charge-controlled-reaction mechanism model [23] and the modified Stern double-layer model [24]. These models generally assume that the adsorption of water reducers follows the Langmuir isotherm equation. The trend, degree, and driving force of the adsorption process can be explained by adsorption thermodynamics, which can also be used to explain the adsorption characteristics, laws, and mechanisms. Stefanie Anne Weckwerth et al. studied the competitive adsorption of PCE with different charge densities and found that high-charge PCE was more easily adsorbed than low-charge PCE, and the affinity and competitiveness of PCE were described as the result of enthalpy and entropy contributions [25]. Adsorption thermodynamic models can relate the molecular structure of PCE, and enthalpy can be well described as linearly proportional to the number of charges per molecule, while entropy effects may play a more important role in directly adsorbing PCE on cement than in exchanging between PCE molecules.

PCE adsorbs onto cement particles, altering the charge density of the cement particles. When the particles approach each other, they repel each other and disperse due to electrostatic forces, while the steric hindrance from the long side chains maintains the dispersion of cement particles. The amount of free water released by PCE dispersion, as well as the morphology and thickness of the PCE-cement particle interfacial layer, have a significant impact on the subsequent cement hydration process. The heat released during hydration varies at different stages of the hydration process, with mineral dissolution, nucleation, and growth of hydration products being the main sources of heat release. The interaction between PCE and cement particles at the early stage of hydration, which is influenced by their different microstructures, results in different rates of mineral dissolution, pore solution concentration, nucleation, and growth rates of hydration products. It is generally believed that the thickness of the new surface film layer caused by adsorption may be thicker than the original layer. The presence of a thick film layer can hinder the exchange of substances and energy between the surface of cement particles and the liquid phase, as well as the penetration of external ions into the interior of cement particles. Hongwei Tian et al. studied the effects of PCE with high charge density on the flowability and early hydration of Portland cement and sulfate-aluminate cement systems and found that it can significantly inhibit the early formation of AFt crystals in SAC paste [26].

The purpose of hydration heat research is to study the heat-release behavior and initial and final states of different hydration processes, but it lacks the main factors that cause heat-release behavior at different stages. Hydration kinetics can characterize the state and reaction process when different reaction mechanisms dominate during the hydration process. By studying the changes in internal and external variables such as cement dissolution rate, ion concentration, nucleation and crystal growth rate of hydration products, and cement component composition on the reaction rate and direction of hydration, the macroscopic and microscopic mechanisms of cement hydration can be revealed, and the entire process of hydration reaction can be dynamically analyzed [27]. R. Krstulovic and P. Dabic established a hydration kinetics model by deducing the total heat of hydration [28]. When the heat of cement hydration enters the acceleration phase, the heat-release activity becomes active. The Krstulovic–Dabic model is currently widely recognized and used in the study of composite cementitious systems [29,30,31]. The Krstulovic–Dabic model [28] suggests that the cement hydration process can be divided into three stages: nucleation and crystal growth (NG), interfacial reaction (I), and diffusion (D), with the NG and I stages being in the acceleration phase when the hydration products are still undergoing rapid growth.

Krstulovic and Dabic characterized the three hydration processes using the JMAK, Brown, and Jander models [28]. The kinetic equations for the three processes are as follows:(1)$${\left[-\ln\left(1-\alpha \right)\right]}^{\frac{1}{n}}=K_{NG}\left(t-t_{0}\right),$$
(2)$${\left[{1-\left(1-\alpha \right)}^{1/3}\right]}^{1}=K_{I}\left(t-t_{0}\right),$$
(3)$${\left[{1-\left(1-\alpha \right)}^{1/3}\right]}^{2}=K_{D}\left(t-t_{0}\right),$$
where α is the degree of hydration at time t, n is the geometric crystal growth index in J·g^−1^, t is the hydration time in hours, $t_{0}$ is the end time of the induction period in hours, and $K_{\mathrm{NG}}$, $K_{I}$ and $K_{D}$ are the reaction rate constants for NG, I, and D processes, respectively.

Hydration kinetics models have been applied in various aspects, such as examining the crystal growth process n value and apparent activation energy change in multiphase systems [32], as well as the dominant reactions of the hydration process in multiphase systems at different temperatures, which are primarily governed by phase boundary reactions in cement with steel slag content exceeding 50% [33]. Lei B et al. established a single-phase hydration kinetics model of cement, demonstrating that the water-cement ratio accelerates phase boundary reactions but does not affect early nucleation and crystal growth [34]. Temperature can accelerate the hydration process but cannot change the final degree of hydration.

Typically, the reaction parameters of different stages are studied in hydration kinetics, and the main parameters of the NG process are K_NG_ and n, where K_NG_ is the rate constant of the crystallization nucleation and crystal growth in the NG process, reflecting the speed of the hydration reaction. The larger the K_NG_, the easier the reaction, significantly increasing the nucleation rate of C-S-H and Ca(OH)_2_ crystals. As the Ca^2+^ concentration in the solution increases, C-S-H is more likely to reach saturation. Meanwhile, the NG process is affected by the reaction order n, which indicates the degree of concentration affecting the reaction rate [35]. Therefore, the fitting curve of the NG process is a parabola. By studying the relationship between the morphology of hydration products and the reaction rate, Hui Zhang obtained that n can also be expressed as the growth index, where n = 1 corresponds to needle-shaped products, usually Ca(OH)_2_ crystals or AFt, and n = 2 corresponds to plate-shaped or sheet-like products, usually the growth form of C-S-H [32]. At the beginning of the acceleration period, sufficient water supply and high ion undersaturation in the liquid phase limits the hydration products. During the acceleration period, the hydration rate is mainly controlled by the growth of cluster-shaped (floc-shaped) C-S-H [36]. As the hydration process reaches the end of the NG process, the lack of growth space restricts the fast migration ability of ions in the liquid phase, and the nucleation sites decrease. When it reaches the turning point of the I or D process, the nucleation and growth rate are at the maximum point [37]. Therefore, the NG process can investigate the nucleation and growth of hydration products during the acceleration period. Y.R. Zhang et al. [38,39,40] believed that the maximum reaction rate is mainly determined by the total number of nucleation sites in the acceleration period. When there are many nucleation sites, the hydration products can form a dense hydration product coating when their size is small enough to contact each other, indicating a lower degree of cement hydration. The I process is the diffusion across the phase boundary, where a large number of ions passes through the phase boundary between cement particles and hydration products, causing the continuous growth of hydration products. As the hydration products increase, the C-S-H gel produced by hydration gradually precipitates on the surface of the particles and forms a hydration product film, making ion migration difficult. K_I_ is the rate constant of phase boundary reactions, and the smaller the K_I_, the lower the concentration of pore solution ions and the interface area between crystals and pore solution, resulting in a smaller phase boundary reaction rate.

This paper designs five different polycarboxylate superplasticizers with varying carboxyl densities and main chain polymerization degrees. The differences in microstructure are studied using hydration heat and hydration kinetics models to investigate the evolution of cement hydration heat and hydration products over time. The Krstulovic–Dabic model is used to establish a hydration reaction kinetics model for the benchmark cement system based on the isothermal microcalorimetry method. The changes in parameters of the kinetic model under different charge densities and molecular weights are examined, and a relationship function between the degree of reaction hydration and the hydration rate is established. By fitting the hydration process of the benchmark cement system, the impact of PCE on the hydration products and cement hydration mechanism, as well as the influencing factors in different hydration stages, are revealed.

## 2. Materials and Methods

### 2.1. Raw Materials

Isobutene-coupled polyethylene glycol ether (TPEG, molecular weight 2400), JiaHua Chemical Co., Ltd., Quanzhou, China; Acrylic acid (AA), Jiangsu Runhan Chemical Co., Ltd., Jiangyin, China; Ammonium persulfate (APS), Jinan Fengle Chemical Co., Ltd., Jinan, China; Sodium hexametaphosphate (SHP), Jining Shunyida Chemical Technology Co., Ltd., Jining, China.

### 2.2. Synthesis

A series of comb-like polycarboxylate superplasticizers with different carboxyl densities and different main chain polymerization degrees were synthesized by the free radical copolymerization of TPEG and AA. In total, 200 g of TPEG, SHP, and deionized water (DI) were added to a four-necked flask and stirred with a magnetic stirrer until dissolved. Then, APS and 20 g of DI were mixed and stirred until dissolved. AA and 12.5 g of AA were mixed and heated in a 60 °C water bath, and the above solution was uniformly dripped using a peristaltic pump. After dripping, the mixture was incubated for 1 h, cooled to 40 °C, and neutralized to pH 6–7 with 32% NaOH. The synthesis process is shown in Figure 2.

### 2.3. Cement

This study used P. I 42.5 ordinary Portland cement that meets the Chinese standard GB 8076-2008. The chemical and mineral composition, as well as the particle size distribution of the cement, are shown in Table 1, Table 2 and Table 3. The chemical composition analysis of cement was determined according to the Chinese standard GB/176-2017 “Methods for chemical analysis of cement”. The mineral composition of the cement was determined based on the data results in Table 1 according to the Chinese standard GB/T 21372-2008 “Portland cement clinker”. The particle size distribution of the cement was measured using the laser particle size analyzer from Jinan Winner Particle Instrument Stock Co., Ltd., Jinan, China, with the model name Winner 3000.

### 2.4. Experimental Methods

#### 2.4.1. Gel Permeation Chromatography (GPC)

The temperature was maintained at 25 °C, and a 0.1 mol/L NaNO_3_ aqueous solution with a pH of 7 was used as the eluent. Glucan with different molecular weights was used as a calibration standard. PCE was diluted to 5 mg/mL with 0.1 mol/L NaNO_3_ solution. GPC was performed using a Waters 1515 instrument (Waters, Milford, MA, USA) equipped with a differential refractive index detector. Additionally, a multi-detection system (Malvern Viscotek 270 Dual Detector) equipped with viscosity and low-angle laser light scattering detectors was utilized.

#### 2.4.2. Infrared Spectroscopy

A PE Spectrum Two spectrometer (PerkinElmer, Waltham, MA, USA) was used with a scanning range of 4000–400 cm^−1^ and a spectral resolution of 4 cm^−1^.

#### 2.4.3. Charge Density

The charge density of PCE was measured using a particle charge detector (PCD 05, Mütek Analytic, Brunswick, Germany). The sample was first diluted to a concentration of 0.1 wt%, and then a 0.001 M solution of poly(diallyldimethylammonium chloride) was titrated into the sample until the negative charge of PCE was completely neutralized.

#### 2.4.4. Fluidity

The fluidity of cement slurry was measured using the homogeneity test method in GB/T 8077-2012 “Test method for homogeneity of concrete admixtures”. The PCE dosage was 0.2% by mass of cement.

#### 2.4.5. Rheology

Sample preparation: A cement slurry with a water-cement ratio (w/c) of 0.29 was prepared using the method in GB8077-2012. PCE was added to the cement slurry at a dosage of 25 mg/g of cement. Rheology measurements were performed using a rotational viscometer (Rheolab QC, Anton Paar, Graz, Austria). The prepared cement slurry was immediately poured into the cylinder. The rotor diameter was 39 mm, the cylinder diameter was 42 mm, and the height of the rheological laminar flow was 1.5 mm.

Testing process: The instrument was monitored for 60 s to stabilize at 20 °C. The sample was pre-sheared at 5 rad/min for 30 s, and then the shear rate was increased to 0–131 rpm for 1 min, followed by holding at 131 rpm for 18 s. The shear rate was then decreased and increased again from 0 to 131 rpm for 1 min. The experimental temperature and humidity were maintained at 20 ± 2 °C and 60 ± 5%, respectively. Three tests were performed for each cement slurry.

The Bingham model Equation (4) and Herschel–Bulkley model Equation (5) were used to determine the yield stress and viscosity. For shear-thinning mixtures, n < 1, and for shear-thickening mixtures, n > 1.
(4)$$\tau_{1}=\tau_{0}{+\mu }_{0}*\gamma ,$$
(5)τ2=τ0+μ0∗γ^n,
where $\tau_{1}$ and $\tau_{2}$ are the shear stresses (Pa); $\tau_{0}$ is the yield stress (Pa); $\mu_{0}$ is the plastic viscosity (mPa·s); γ is the shear rate (s^−1^); and n is the rheological behavior index.

#### 2.4.6. Total Organic Carbon (TOC)

The adsorption amount was measured using an Alemont TOC−VCPH instrument (Alemont GmbH, Berlin, Germany). Different concentrations of PCE solution were prepared, and 20 g of Portland cement was added to 40 mL of the PCE solution. After thorough mixing, an appropriate amount of the liquid was transferred to a centrifuge tube and centrifuged for 10 min at 5000 r/min. The supernatant was collected for TOC testing, and the adsorption amount of PCE on cement particles was calculated using Equation (6):
(6)$$\Gamma \;=\;\frac{(c_{1}-c_{0})\times V_{1}}{m},$$
where Γ is the adsorption amount of PCE in the cement slurry (mg/g); $c_{0}$ is the total organic carbon content in the PCE solution added to the cement slurry (mg/L); $c_{1}$ is the total organic carbon content in the admixture sample solution (mg/L); $V_{1}$ is the volume of the admixture sample solution added in the experiment (mL); and m is the mass of cement (mg).

#### 2.4.7. Heat of Hydration

The hydration heat curve of cement was measured using a TA Instruments TAM Air isothermal microcalorimeter (TA Instruments, New Castle, DE, USA). Before the experiment, PCE, Portland cement, and DI water were placed in a 25 °C constant temperature room for 24 h to minimize experimental errors. A water-to-cement ratio of 0.29 was used. Then, 100 g of Portland cement and 29 g of PCE aqueous solution (PCE solid content was 0.2% by mass of cement) were mixed thoroughly, and 3.0000 g of the cement paste was taken for 72 h of hydration heat testing.

#### 2.4.8. SEM

A cement slurry was prepared with a PCE concentration of 0.20 wt% (by mass of cement) in DI water with a water-to-cement ratio of 0.29. The morphology of the hydration products was determined using an environmental scanning electron microscope (SEM) with a resolution of 100,000 times. The SEM used was a Korean COXEM-20 (COXEM Co., Ltd., Daejeon, Republic of Korea).

## 3. Results and Discussion

### 3.1. Gel Permeation Chromatography

To investigate the effect of different carboxyl group densities on the performance of PCE, the AA to TPEG molar ratio was designed to be 1.5:1 to 6:1, APS was used as the initiator and SHP was used as the chain transfer agent. The amount of initiator was 0.5% of the total amount of TPEG, and the chain transfer agent was from 0.5% to 2.0% of the total amount of TPEG.

The molecular structure of the synthesized PCE was characterized using gel permeation chromatography, and the results are shown in Table 4.

The calculation formulas for the side chain density $\rho_{s}$ and the main chain degree of polymerization $L_{m}$ are shown in Equations (7) and (8), respectively.
(7)$$\rho_{s}=1-\frac{n\left[AA\right]}{\begin{array}{c} n\left[AA\right]+n\left[TPEG\right]*Z_{z} \end{array}},$$
where $Z_{Z}$ represents the conversion rate of the TPEG monomer, and n[TPEG] and n[AA] represent the amounts of TPEG and AA monomers added during the polymerization reaction, respectively.
(8)$$L_{m}=\frac{M_{w}}{\begin{array}{c} M \end{array}},$$
where $M_{w}$ is the weight-average molecular weight of the PCE, and M is the theoretical molecular weight of the structural unit.

Table 4 presents the molecular structure parameters of the PCE samples with different carboxyl group densities. The side chain density decreased gradually from 0.31 to 0.13 as the carboxyl group density increased from 1.5 to 6, and the trends of the side chain density and the carboxyl group density were inversely proportional. The molecular weight data in Table 4 show that as the carboxyl group density increased, the number-average molecular weight (Mn) and the weight-average molecular weight (Mw) gradually increased. Specifically, compared to PCE-1, the Mn and Mw of PCE-4 increased by approximately 1.2 and 1.3 times, respectively, mainly because the number of acrylic acid units and the number of TPEG side chains increased with the carboxyl group density, increasing the Mn and Mw. The Mn and Mw of PCE-2 increased significantly compared to PCE-1, possibly due to the lower conversion rate of PCE-1 and the increased charge density. For the samples with different main chain degrees of polymerization, as the amount of the chain transfer agent increased, Mn and Mw gradually decreased. Specifically, compared to PCE-5, the Mn and Mw of PCE-7 decreased by approximately 2.5 and 3.0 times, respectively, mainly because the increased amount of the chain transfer agent led to more termination reactions and a decrease in the degree of polymerization, decreasing the Mn and Mw. Meanwhile, the charge density decreased.

The α value in the Mark–Houwink equation is considered to be a coefficient related to the shape of the polymer chain and reflects the conformation of the polymer in the solvent. The α values in Table 3 are all less than 0.5, indicating that the polymer exhibited a curled spherical structure in the sodium nitrate mobile phase. As the carboxyl group density gradually increased, the fluid mechanics radius (Rh) of the sample increased due to more calcium ion complexation. As the amount of chain transfer agent increased, the Rh of the sample decreased and α also decreased, indicating an increase in the curling degree and a tendency toward a more compact spherical structure. Refer to Figure 3 for schematic diagrams.

### 3.2. Infrared Spectral Analysis

The synthesized PCE was characterized using an infrared spectrometer, and the results are shown in Figure 4.

As shown in Figure 4, the peak positions of the infrared spectra of PCE with different carboxyl group densities were the same. There was no stretching vibration absorption peak of C=C double bonds in the range from 1600 cm^−1^ to 1680 cm^−1^, indicating that the reaction between the polycarboxylic acid and the water-reducing agent was relatively complete. The absorption peak at 2866 cm^−1^ was due to the stretching vibration of C-H bonds. The absorption peak at 1728 cm^−1^ was the symmetric stretching absorption peak of the C=O bond in the carboxyl-COOH group. When the acid ether ratio was 1.5, the amount of acrylic acid was low, and the absorption peak here was weak. The absorption peak at 1350 cm^−1^ is the characteristic absorption peak of the side chain polyethylene glycol (CH_2_CH_2_O)_n_, and the strongest absorption peak at 1107 cm^−1^ was caused by the asymmetric stretching vibration of the ether bond C-O-C. From the infrared spectrum, it can be seen that the polymers all had carboxyl and polyethylene glycol-related groups, so it can be inferred that the samples were all binary copolymers of acrylic acid and TPEG monomer.

### 3.3. Charge Density

The PCE charge density was tested using a particle charge meter, and the results are shown in Figure 5.

As shown in Figure 5, the order of the PCE charge density was PCE-4 > PCE-5 > PCE-6 > PCE-3 > PCE-2 > PCE-7 > PCE-1, indicating that as the carboxyl group density increased, the charge density of PCE increased, and as the main chain polymerization degree decreased, the charge density of PCE decreased. Among them, when the carboxyl group density was 6.0, the maximum number of carboxyl groups per unit length of chain segment and the highest degree of ionization resulted in the highest charge density.

### 3.4. Dispersion Performance

#### 3.4.1. Fluidity

The effect of different microstructure PCE on the flow performance of cement slurries was studied, and the experimental results are shown in Figure 6. Under the same dosage, with the increase of carboxyl density or main chain polymerization degree of PCE, the initial flowability of the slurry first increased and then decreased. Under different dosage conditions, with the increase of PCE dosage, the initial flowability of the slurry gradually increased. It can be seen that with the increase of the main chain carboxyl density, the charge density increased, and the adsorption capacity and dispersion performance improved. After the carboxyl density of the main chain increased to a certain amount, the carboxyl groups of the water-reducing agent bridge were adsorbed onto different cement particles and complexed with calcium ions in the solution, resulting in a decrease in the dispersion performance. Furthermore, combining the PCE molecular structural parameter analysis in Table 4, when the side chain density was 0.31, the charge density was lower, and the dispersion performance was poor and non-flowable when the dosage was between 0.2% and 0.3%. When the main chain polymerization degree was too large, the dispersion performance also decreased. Due to the excessively long main chain, the molecular chains of the water-reducing agent entangled with each other. Although the spatial hindrance of PCE was large, the main chain carboxyl groups were wrapped inside and could not be adsorbed onto the surface of the cement particles, resulting in a smaller dispersion performance. When the main chain polymerization degree reached a certain degree, the molecular chains of PCE became more extended, and the carboxyl groups on the main chain were fully adsorbed onto the surface of the cement particles. Therefore, the dispersion performance of the cement slurry gradually increased. At the same time, with the continued decrease in the main chain polymerization degree, the spatial hindrance decreased, the carboxyl density decreased, the adsorption amount of the polycarboxylic acid molecule on the surface of the cement particles showed a decrease, and the dispersion performance decreased.

#### 3.4.2. Rheology

The internal structural characteristics of cement slurries can be characterized by the rheological properties of the slurry. This paper studied the effect of different carboxyl densities and main chain polymerization degrees on the rheological properties of cement slurries by analyzing the changes in the flowability, plastic viscosity, and yield stress of the slurry. The rheological property data are shown in Table 5. The shear rate and shear strain curves under different conditions are shown in Figure 7.

When the dosage was 0.2%, with the increase of carboxyl density, the initial flowability of the slurry first increased and then decreased, and the plastic viscosity gradually decreased. At the same time, the yield stress first decreased and then increased, while the thixotropy area was the smallest for PCE-3. With the increase of the main chain polymerization degree, the yield stress of PCE-6 and PCE-3 was smaller, and the thixotropy area was smaller, which is consistent with the conclusion that the flowability was larger.

### 3.5. Adsorption Performance

#### 3.5.1. Adsorption Isotherm Equation

The adsorption amount of different PCE structures on cement particle surfaces was tested under different PCE dosages, as shown in Figure 8.

Under different dosage conditions, the adsorption amount of PCE samples gradually increased with the increase of carboxyl group density. In the lower dosage range (<0.5 mg·L^−1^), the adsorption amount of PCE1~PCE-4 increased sharply. However, there was a significant difference in the saturation adsorption amount of the four polymers. At high dosages (>1 mg·L^−1^), the adsorption amount of polymers PCE-1 and PCE-2 continued to increase and gradually reached a steady-state value (0.5~1.2 mg·g^−1^). However, the adsorption amounts of polymers PCE-3 and PCE-4 still increased sharply until they approached their steady-state value of 1.7~2.0 mg·g^−1^, which was significantly higher than the steady-state values of PCE-1 and PCE-2. The reason for this was the difference in solution conformation. As the carboxyl group density increased, both the hydrodynamic radius and charge density increased, increasing the adsorption amount. This also confirms the conclusion of the previous molecular structure.

From Figure 8, it can be seen that when the main chain degree of polymerization was between 16.46 and 22.29, the adsorption amount increased with the decrease of the main chain degree of polymerization. This was because as the PCE molecular weight decreased, the hydrodynamic radius Rh decreased (molecular volume decreased), the charge density decreased, and the Mark–Houwink α decreased (curvature degree increased). Although the number of anchor points that can undergo adsorption decreased, the reduction in molecular volume and the increase in curvature degree reduced the space occupied by a single PCE molecule, allowing more PCE to adsorb in the same space. When the main chain degree of polymerization was between 11.18 and 16.46, the adsorption amount decreased with the decrease of the main chain degree of polymerization. This was because the increase in adsorption amount caused by the reduction in molecular volume was lower than the decrease in adsorption amount caused by the reduction of -COO- groups, resulting in an overall decrease in the adsorption amount.

To further determine the characteristic adsorption platform of PCE, the differences in the adsorption performance of different PCE samples were compared by fitting the Langmuir, Freundlich, Temkin, and R-P adsorption models, as shown in Table 6.

From Table 6, it can be seen that PCE-1, PCE-2, and PCE-7 had the highest correlation coefficients (R^2^) fitted by the Langmuir adsorption model, indicating that when the carboxyl group density was low or the main chain polymerization degree was high, their adsorption on the surface of cement particles was mainly characterized by Langmuir adsorption features. A large amount of research has shown that the driving forces for the adsorption of PCE water reducers on cement particle surfaces are mainly electrostatic interactions and chelation coordination. The saturated adsorption amount of PCE-1 on cement increased from 0.8295 mg/g to 1.4175 mg/g for PCE-2, and K_L_ decreased from 1.8620 to 0.8915, indicating that an increase in carboxyl group density is favorable for PCE adsorption on cement. However, with the decrease of the main chain polymerization degree, there was no apparent regularity in the saturated adsorption amount q_e_ on cement. However, K_L_ first decreased and then increased, indicating that both too-high and too-low main chain polymerization degrees are unfavorable for adsorption. R-P model fitting of PCE-1, PCE-2, PCE-3, and PCE-5 also had relatively large correlation coefficients, indicating that samples with smaller carboxyl group densities or higher main chain polymerization degrees also conformed to R-P adsorption characteristics. With an increase in carboxyl group density, k_RP_ gradually decreased, indicating a lower degree of adsorption, which may be related to the fact that PCE-2 had the smallest saturated adsorption amount and reached saturation more quickly. PCE-1, PCE-4, PCE-5, PCE-6, and PCE-7 also had relatively large R^2^ values when fitted to the Temkin adsorption model, indicating that their adsorption on the surface of cement particles also conformed to the Temkin adsorption characteristics. k_T_ is the Temkin constant, reflecting the size of the adsorption between the adsorbate and adsorbent, and a larger k_T_ indicates a stronger interaction. As the carboxyl group density increased or the main chain polymerization degree decreased, k_T_ gradually decreased, indicating a weakening of the interaction. b_T_ represents the change in adsorption energy and reflects the type of adsorption, and a smaller b_T_ tends toward physical adsorption, while a larger b_T_ tends toward chemical adsorption. With an increase in carboxyl group density, b_T_ decreased from 15,494.73 to 3183.23, indicating a trend from chemical adsorption to physical adsorption. With a decrease in the main chain polymerization degree, b_T_ first decreased and then increased, indicating a transition from chemical adsorption to physical adsorption and then back to chemical adsorption. PCE-2, PCE-5, and PCE-6 had relatively large R^2^ values when fitted to the Freundlich adsorption model, indicating that their adsorption on the surface of cement particles conformed to the Freundlich adsorption characteristics. With an increase in carboxyl group density, K_F_ increased, adsorption capacity increased, and n_F_ decreased, but both were greater than 1, indicating single-layer adsorption. With a decrease in the main chain polymerization degree, K_F_ first increased and then decreased, and n_F_ first decreased and then increased, but both were greater than 1, indicating single-layer adsorption.

#### 3.5.2. Adsorption Thermodynamics

The Langmuir adsorption model was used to fit the adsorption isotherms of PCE on the surface of cement particles at temperatures of 298 K, 308 K, 318 K, and 328 K. The Langmuir equilibrium constant $K_{L}$ (L·mg^−1^) reflects the adsorption affinity and can be used to calculate the adsorption free energy ∆G, adsorption enthalpy ∆H, and adsorption entropy ∆S, as shown in Equations (9)–(11). The parameters of the Langmuir equation for the adsorption of PCE on cement particles are shown in Table 7.
(9)$$R_{L}=\frac{1}{1+K_{L}C_{0}},$$
(10)$$∆G=-RTlnK_{L},$$
(11)$$lnK_{L}=-\frac{∆H}{R}\cdot \frac{1}{T}+\frac{∆S}{R},$$
where R is the ideal gas constant (8.314 J·mol^−1^·K^−1^); R is the absolute temperature; $R_{L}$ (dimensionless) stands for separation factor; and $C_{0}$ (mg·L^−1^) is initial concentration of the adsorbate in the solution.

As shown in Table 7, within the experimental temperature range from 298 K to 328 K, ΔG was negative at each temperature, indicating that the adsorption of PCE on cement particles is a spontaneous process. As the temperature increased, the Langmuir equilibrium constant K_L_ decreased, and the separation factor R_L_ increased, indicating that the increase in temperature is unfavorable for the adsorption of PCE on cement particles. The adsorption entropy changes ΔS of PCE were greater than zero, indicating that the adsorption process is entropy-driven, and the increase in entropy caused by the desorption of adsorbate molecules or ions exceeds the decrease in entropy caused by the adsorption of solute molecules. The adsorption enthalpy changes ΔH of PCE were negative, indicating that the adsorption process on the cement surface is exothermic. As the temperature decreased, it was conducive to the adsorption process, and the absolute value of the enthalpy change (∆H < 37.62 kJ·mol^−1^ and >11.99 kJ·mol^−1^) was small, indicating that the adsorption process has both physical and chemical adsorption characteristics. The value of ΔG was between −29.97 and −21.47, and the dominant forces were electrostatic adsorption and hydrogen bonding. It is speculated that the adsorption of PCE on cement particles is mainly a physical adsorption process, with chemical adsorption as a secondary effect.

#### 3.5.3. Adsorption Kinetics

After the addition of PCE, it adsorbed on the surface of the cement. Studying the adsorption rate of PCE helps us to investigate the competition between cement and polycarboxylate superplasticizers. When the PCE dosage was 4 mg·L^−1^, the adsorption amount of different structured PCE at different adsorption times was tested, as shown in Figure 9.

Figure 9 shows that different PCE exhibited similar regularity in the adsorption kinetics on the surface of cement particles. The adsorption amount of PCE on the surface of cement particles increased with the extension of adsorption time and reached adsorption equilibrium at about 60 min, where a plateau appeared. With the increase of side chain density, the saturated adsorption amount gradually increased. At the beginning of adsorption, PCE was mainly adsorbed on the outer surface of cement particles, and the adsorption rate was fast. At this time, adsorption was mainly controlled by diffusion. As adsorption continued, some PCE was wrapped by hydration products, resulting in a gradual decrease in concentration. At this time, adsorption was mainly controlled by the chemical adsorption process, and the adsorption rate gradually slowed down. In the late stage of adsorption, as the driving force of concentration difference became smaller and smaller, adsorption reached equilibrium.

To more clearly indicate the influence of PCE on the adsorption rate, quasi-first-order adsorption kinetics (PFO), quasi-second-order adsorption kinetics (PSO), Elovich, and intraparticle diffusion (ID) adsorption kinetics equations were studied, and the results are shown in Table 8.

The fitting results of Table 8 show that the PSO and Elovich models had the best fitting effect for the adsorption kinetics equation of different carboxyl group densities and main chain polymerization degree PCE. The PSO adsorption equation is based on the assumption that the adsorption rate is controlled by the chemical adsorption mechanism and involves composite adsorption reactions including external liquid film diffusion, surface adsorption, and intraparticle diffusion. The Elovich adsorption equation describes non-uniform diffusion processes controlled by reaction rates and diffusion factors. The correlation coefficients of the two models’ fittings were between 0.9568 and 0.9958, indicating a relatively high fitting correlation. As the carboxyl group density increased, the equilibrium adsorption amount q_e2_ increased, the quasi-second-order adsorption rate constant k_2_ decreased, and the initial adsorption rate h_0_ increased first and then decreased. This indicates that with the progress of the hydration reaction, more hydration products are produced, providing more adsorption sites, which is beneficial to the adsorption of carboxyl groups. However, the maximum value of q_e2_ of PCE-2 was not the largest, indicating that the equilibrium adsorption amount may be related to the chemical reaction between PCE and hydration products with the increase of carboxyl group density. As the main chain polymerization degree decreased, q_e2_ changed slightly, k_2_ increased, and h_0_ increased. This indicates that the smaller the main chain polymerization degree, the faster PCE diffuses through the external liquid film, which is related to the PCE conformation, and the smaller the molecular weight, the smaller the hydrodynamic radius, the easier it is to diffuse and adsorb. The smaller the main chain polymerization degree, the faster the adsorption rate, and the lower the charge density, the higher the equilibrium adsorption amount.

With the increase of carboxyl group density, the initial adsorption rate first increased and then decreased, and the analytical constant b gradually decreased. This indicates that the adsorption rate is higher when the carboxyl group density is in the middle, which is consistent with the conclusion of the PSO adsorption equation. The greater the carboxyl group density, the stronger the adsorption capacity and the less likely it is to desorb, resulting in a smaller b value. As the main chain polymerization degree decreased, both a and b gradually increased, indicating that the larger the main chain polymerization degree, the more adsorption sites there are and the less likely it is to desorb. However, the initial adsorption rate was also the highest for PCE-7, which is consistent with the conclusion of the PSO adsorption equation.

In summary, both PSO and Elovich can well describe the adsorption kinetics of different microstructure PCE, and their conclusions are consistent.

### 3.6. Cement hydration

#### 3.6.1. Heat of Hydration

The exothermic process of cement hydration usually consists of five stages [41], as shown in Figure 10.

The heat flow and cumulative heat curves of hydration are shown in Figure 11. The characteristic parameters extracted from the heat curves are listed in Table 9.

As shown in Figure 11, compared to the blank, the addition of PCE delayed the induction period of hydration, and the reason for this is that PCE hindered the diffusion of mineral ions into the solution [42], resulting in a slower dissolution rate of minerals, slow growth of solution ion concentration, and delayed nucleation of C-S-H hydration products. With the increase of carboxyl density, the induction period t_0_ increased, which was consistent with the trend of PCE adsorption. With the increase of the main chain polymerization degree, t_0_ first increased and then decreased. PCE-7 had the largest main chain polymerization degree and may have absorbed adjacent cement particles, resulting in reduced dispersion, less released free water, and requiring less solution volume to achieve Ca^2+^ supersaturation. Therefore, the induction period was shorter.

During the acceleration period, the hydration reaction rate was fast, the Ca^2+^ supersaturation decreased, and it was in the stage of nucleation and growth of hydration products [43]. According to Table 9, the acceleration period t_3_-t_0_ of PCE was 30% longer than that of the blank sample, indicating that PCE inhibits cement hydration during the acceleration period. Among the samples with different carboxyl densities, PCE-4 had the shortest acceleration period. According to the dispersibility in Table 5, the supersaturated solution volume of PCE-4 was smaller than that of PCE-3, but PCE-4 had a higher carboxyl density, which can chelate more calcium ions, and the residual poly-carboxylic acid molecules in the pore solution were more easily adsorbed on the surface of the hydration products. For PCE samples with different main chain polymerization degrees, PCE-5 had the smallest supersaturated solution volume, but its molecular weight was the smallest, and the steric hindrance of the residual PCE in the solution was smaller, resulting in less dispersion of hydration products, slower nucleation and growth, and a longer duration of the acceleration period.

The maximum slope K_2_ of the acceleration period reflects the nucleation rate of C-S-H in the early stage of the acceleration period [42]. With the increase of carboxyl density, K_2_ gradually increased, promoting the nucleation rate of the first half of the acceleration period. With the decrease of the main chain polymerization degree, K_2_ changed insignificantly.

At the maximum heat release peak, as the carboxyl group density increased, q_3_ first increased and then decreased, and PCE-3 reached its maximum. Meanwhile, q_3_ increased with the increase of the main chain polymerization degree, resulting in consistent heat release rates between PCE-3 and PCE-7, but the heat release amount of the latter was greater than the former. PCE-7 had the longest duration in the acceleration period, resulting in the highest heat release amount; hence, PCE-3 hydrates faster. In summary, compared with others, PCE-3 promotes the hydration of cement in the acceleration period.

#### 3.6.2. Hydration Kinetics

The hydration reaction α-dα/dt curves of the samples are shown in Figure 12, and the summary of the hydration kinetic parameters is shown in Table 10.

From Table 10, it can be seen that in the nucleation and growth process, K_NG_ can accurately determine the change of nucleation points before and after crystal growth, thereby affecting the change of nucleation amount. As the carboxyl group density increased, the crystallization nucleation and crystal growth rate constants K_NG_ gradually increased, and the C-S-H growth volume increased. The reaction order n gradually decreased, with PCE-4 having the smallest n value, and its ion concentration had the least impact on nucleation and growth rates [30]. The crystal growth rate was faster, the single crystal volume was larger, and it occupied nucleation sites and space, so PCE-4 had the smallest nucleation amount when the induction period ended. As the main chain polymerization degree decreased, K_NG_ and n did not change significantly. In addition, the numerical range of n corresponded to the different morphologies of the hydration products, where n was between 1 and 2 for needle-shaped and sheet-like hydration products. The analysis results of K_NG_ and K_2_ have the same trend, and the nucleation amount was inferred consistently.

In the induction period, as the carboxyl group density increased, K_I_ gradually increased, and the interfacial area between crystals and pore solution increased, leading to an increase in C-S-H gel. As the main chain polymerization degree increased, K_I_ did not vary significantly, and it was inferred that PCE with different molecular weights has little effect on the amount of C-S-H gel.

From Figure 12, it can be seen that the deceleration period and the stable period were well-fitted, and the deceleration period was slightly smaller than the actual hydration rate, but it still reflected the hydration reaction process well. K_D_ was almost one order of magnitude lower than K_I_, demonstrating that interparticle diffusion does not require liquid participation. In the deceleration period, the K_D_ blank and PCE sample were consistent, indicating that PCE has a small influence on the D process.

#### 3.6.3. Hydration Products

Using a water-to-cement ratio of 0.35, the morphology features and corresponding hydration degrees of hydration products with different carboxyl group densities and different main chain polymerization degrees were studied by analyzing the morphology features of different hydration times, as shown in Table 11.

Comparing the hydration degree of different samples at the same hydration time can measure the morphology of the hydration products of samples at that time point. As shown in Table 11, the hydration degree of PCE-4 was the highest at the hydration time of 2 h except for the blank sample. The hydration degree of the blank sample was only 0.01, indicating that the hydration products or mineral dissolution heat was almost non-existent.

From Table 12, it can be seen that after 2 h of hydration, the surface of the PCE sample was relatively smooth, with corrosion pits left by mineral dissolution. According to Table 12, the hydration degree of the PCE sample was lower than that of the blank, and the surface of the cement sample was still smooth without any hydration products.

After 12 h of hydration, the blank sample and the PCE sample were, respectively, at the beginning of the D process and I process. The hydration degree of the blank sample was twice that of the PCE sample, and the needle-shaped crystals were more obvious on the surface of the blank sample, while the PCE sample had few or no needle-shaped crystals. The reason for this is that there was less boundary contact of hydration products on the PCE sample, and there were activation sites on the cement particles that provided a platform for the growth of hydration products. At this time, the activation energy for particle growth was lower than that for needle-shaped crystal nucleation, so the PCE sample was still in the rapid nucleation stage.

From Table 13, after 1 day of hydration, according to Table 11 and Figure 12, the hydration degree of the PCE sample was slightly lower than that of the blank sample, and both entered the D stage, indicating that PCE delays the mid-to-late stage of cement hydration to some extent, but the impact is relatively small. With the increase of carboxyl group density, the hydration products of PCE-1 formed cluster-like fluffy structures, while the hydration products of other samples were needle-shaped, with a length of about 1μm. With the increase of main chain polymerization degree, the surfaces of PCE-5 and PCE-6 cement particles were covered with many grains of different orientations, and there were fewer pores and fiber-shaped hydration products of Ca(OH)_2_, while PCE-3 and PCE-7 had more fiber-shaped and porous hydration products, indicating that the hydration degree gradually decreased as the main chain polymerization degree decreased. As shown in Table 12, for PCE-4 samples with different carboxyl group densities, the number of cluster-like C-S-H and needle-shaped and blocky AFt was relatively small, but the needle-shaped crystals were more obvious and showed a biased needle-like shape, which is consistent with the analysis results of $K_{NG}$ and n values. The number of needle-shaped crystals were fewer for samples with different main chain polymerization degrees, and the hydration products tended to be cluster-like, needle-shaped, and blocky, which is consistent with the analysis results of the n values.

Based on Table 11, it can be seen that the hydration degree of PCE samples was slightly higher than that of blank samples, indicating that PCE slightly promotes later hydration and is beneficial for strength improvement. As shown in Figure 12, both blank and PCE samples entered the D process. The morphology of hydration products in PCE samples did not differ significantly and was characterized by a gel matrix with some voids, with needle-shaped, rod-shaped, and plate-shaped hydration products.

#### 3.6.4. Schematic Diagram of Hydration

Figure 13 shows the hydration process during the three dominant reaction periods, which can be roughly inferred from the hydration kinetics and morphology analysis. During the NG process, the layer of hydration products on the surface of particles was relatively thin, and the hydration products (C-S-H and CH) nucleated on and near the surface of hydrated particles, with a small amount of AFt existing in the solution. During the I process, the water-reducing agent originally present on the surface of cement particles was wrapped by the hydration layer and lost its effect, while a small amount of water-reducing agent in the pore solution was adsorbed on the surface of hydration products, providing more space for the growth of hydration products. The layer of hydration products gradually thickened, and some of the hydration products came into contact with each other. During the D process, some needle-shaped hydration products were wrapped by amorphous C-S-H, and the layer of hydration products spread inward to the cement particles. The consumption of Ca^2+^ and sulfate ions in the pore solution was almost exhausted, and the hydration products were in close contact, with the slowest diffusion rate. As the porosity decreased, the strength gradually increased. At the same time, as the consumption of sulfate ions occurred, monosulfate hydrate calcium aluminate (AFm) gradually emerged, and the fibrous AFt phase underwent a transformation into hexagonal plate-like AFm phase.

## 4. Conclusions

Based on the study of the microstructure, rheology, hydration heat, and hydration kinetics of PCE with different carboxyl group densities and different main chain polymerization degrees (PCE), the following conclusions were drawn:With the increase of carboxyl group density, the molecular weight, conversion rate, and hydrodynamic radius increased, while the α value first decreased and then increased. With the increase of the main chain polymerization degree, the molecular weight increased rapidly, and the conversion rate remained at around 90%. This indicates that carboxyl group density has a greater impact on the conversion rate during synthesis.With the increase of carboxyl group density and main chain polymerization degree, the fluidity of the paste showed a trend of first increasing and then decreasing, consistent with the rheological parameters. PCE-3 had optimal dispersion and rheological properties.The adsorption amount increased with the increase in dosage until it reached a steady state. When the carboxyl group density was small or the main chain polymerization degree was large, it showed single-molecule layer adsorption, while the adsorption effect weakened when the carboxyl group density was maximum. The main chain polymerization degree was the determining factor for the adsorption rate on cement, and the charge density was the determining factor for the equilibrium adsorption amount. The smaller the main chain polymerization degree, the faster the adsorption rate, and the smaller the charge density, the higher the equilibrium adsorption amount.With the increase of carboxyl group density, the induction period time gradually increased, and the increase slowed down. PCE-4 had the shortest acceleration period. With the increase of the main chain polymerization degree, the induction period time first increased and then decreased, opposite to the acceleration period.Cement hydration kinetics: during the NG stage, the blank sample had the smallest n value and more needle-shaped products were formed, while the addition of PCE samples resulted in a larger n value and domination of C-S-H products, indicating a delay in the early stage of acceleration period. With the increase of carboxyl group density, PCE-4 had the smallest n value, and the nucleation amount was the smallest in the I process. With the increase of the main chain polymerization degree, PCE-7 had the largest n value, indicating that ion concentration had the greatest influence on its nucleation and growth agent.PCE-4 had fewer clustered C-S-H, needle-shaped, and blocky AFt, but more obvious needle-shaped products, with fewer quantities and more inclined toward needle-shaped, which is consistent with the hydration kinetics analysis. In PCE with different main chain polymerization degrees, there were fewer needle-shaped products, and the products tended toward clustered and blocky AFt, consistent with the analysis results of the n value. The addition of PCE had a significant impact on the hydration induction period and acceleration period, improving the hydration degree of 3d hydration samples and benefitting the development of later strength.

## Figures and Tables

**Figure 1 polymers-15-02496-f001:**
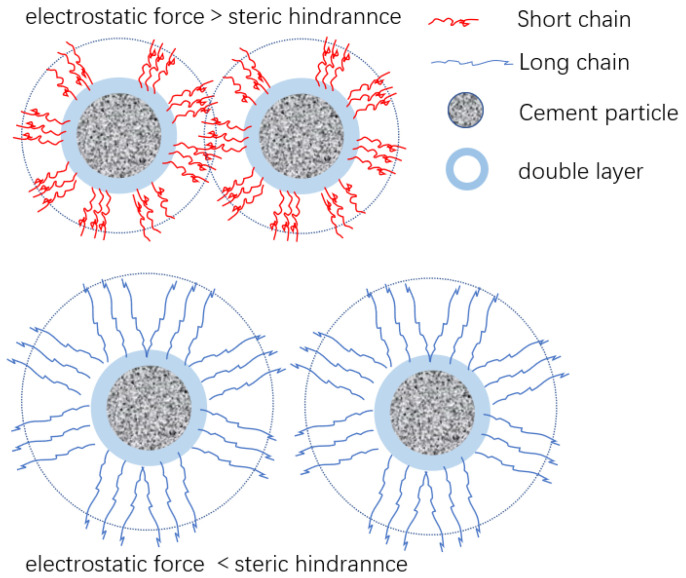
Schematic representation of the adsorption and steric hindrance of PCE on cement particles.

**Figure 2 polymers-15-02496-f002:**
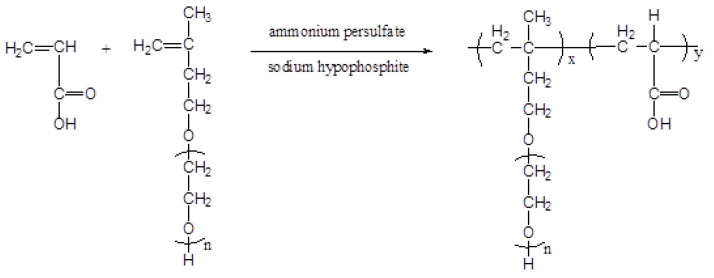
PCE synthesis process.

**Figure 3 polymers-15-02496-f003:**
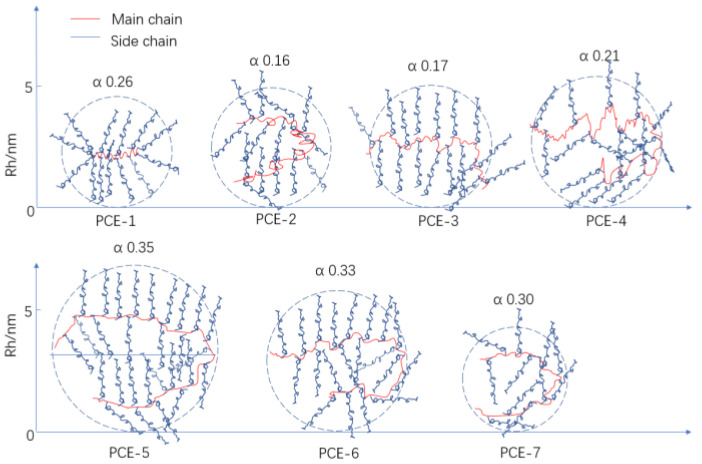
Schematic diagram of the molecular conformation of PCE.

**Figure 4 polymers-15-02496-f004:**
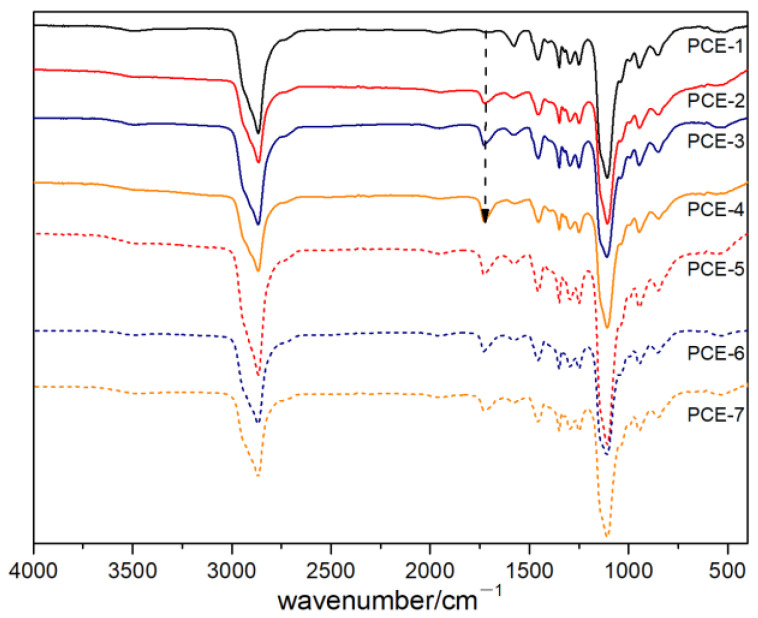
Infrared spectrum of PCE.

**Figure 5 polymers-15-02496-f005:**
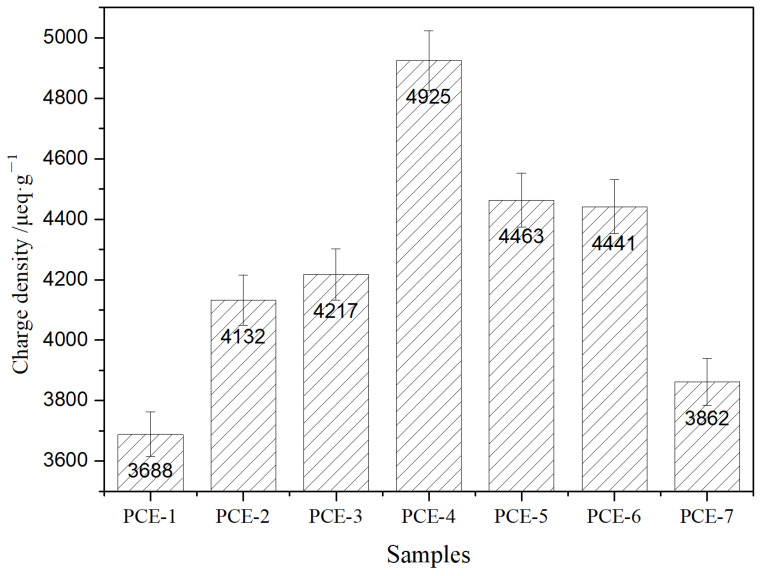
The charge density of different PCE.

**Figure 6 polymers-15-02496-f006:**
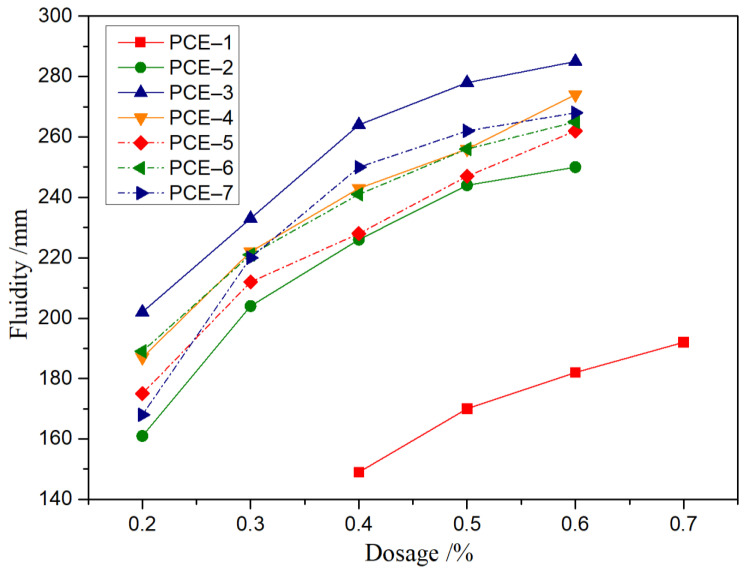
Dispersion performance of PCE with different dosages.

**Figure 7 polymers-15-02496-f007:**
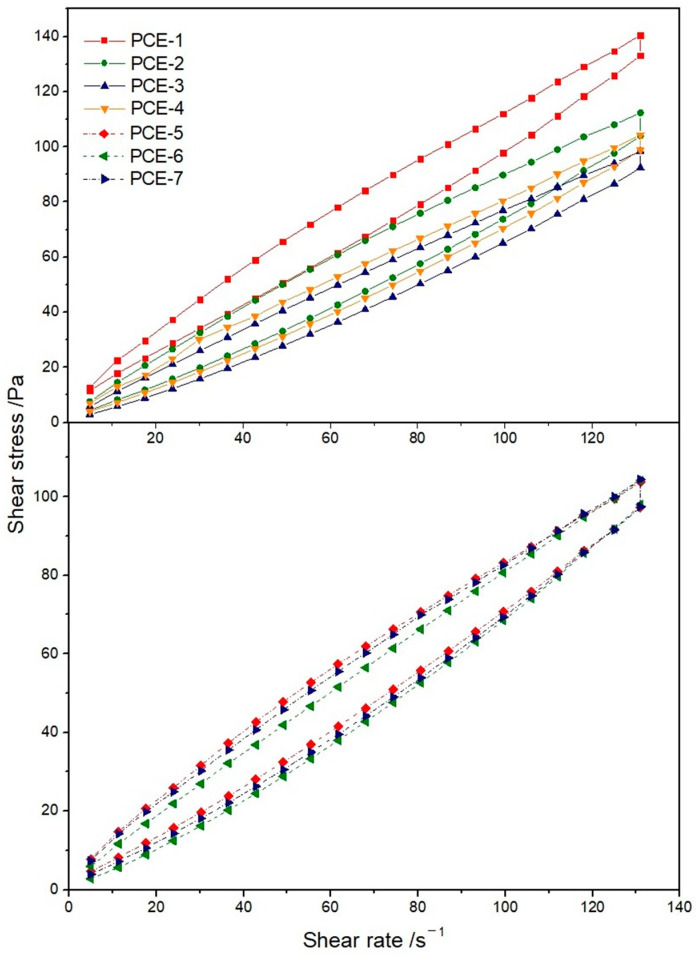
Rheological curves of samples with different structure.

**Figure 8 polymers-15-02496-f008:**
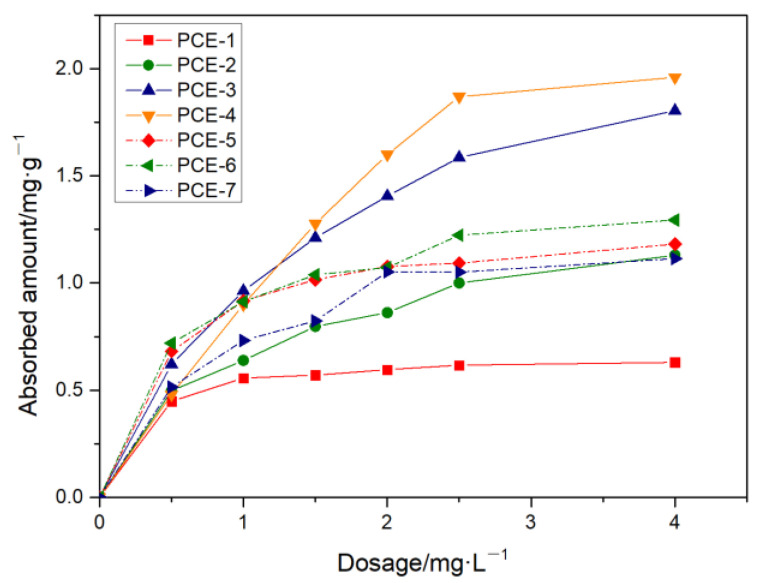
Adsorption amount of PCE with different dosages.

**Figure 9 polymers-15-02496-f009:**
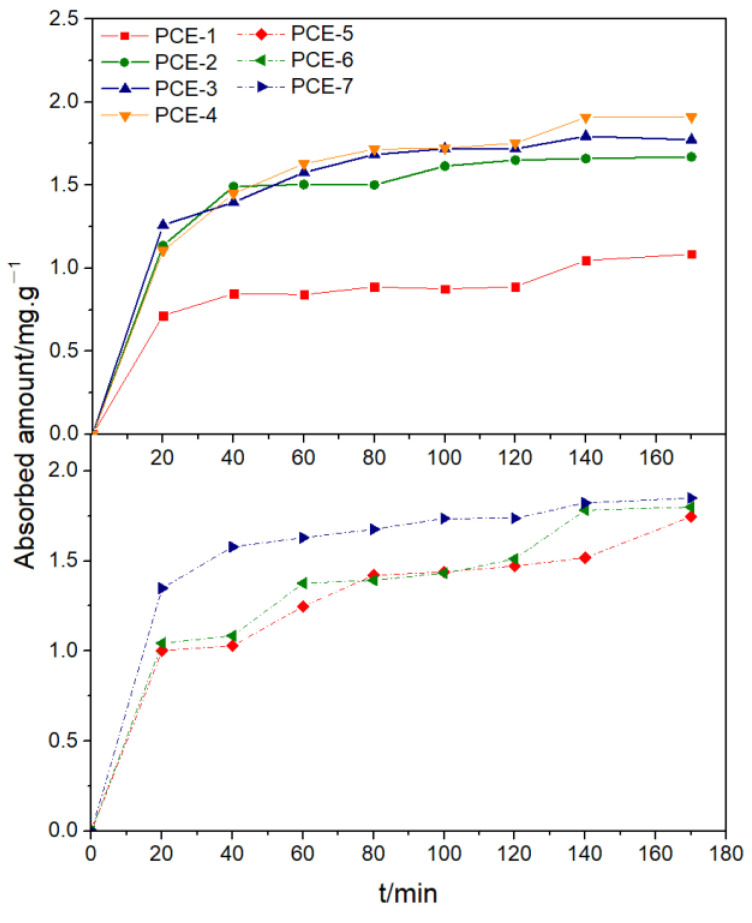
Relationship between adsorption amount and adsorption time of PCE on the surface of cement particles.

**Figure 10 polymers-15-02496-f010:**
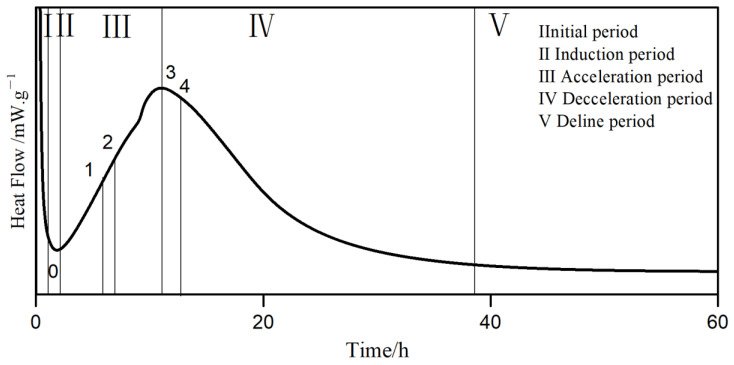
Five stages of heat evolution during cement hydration.

**Figure 11 polymers-15-02496-f011:**
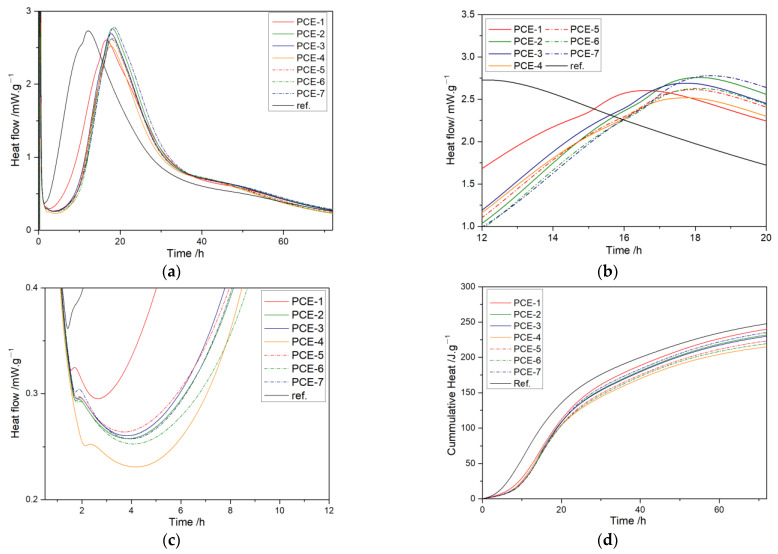
Heat flow (**a**); amplified heat flow curve during the acceleration period (**b**); amplified heat flow curve during the induction period (**c**); cumulative heat (**d**) curves of PCE with different carboxyl densities and main chain polymerization degrees.

**Figure 12 polymers-15-02496-f012:**
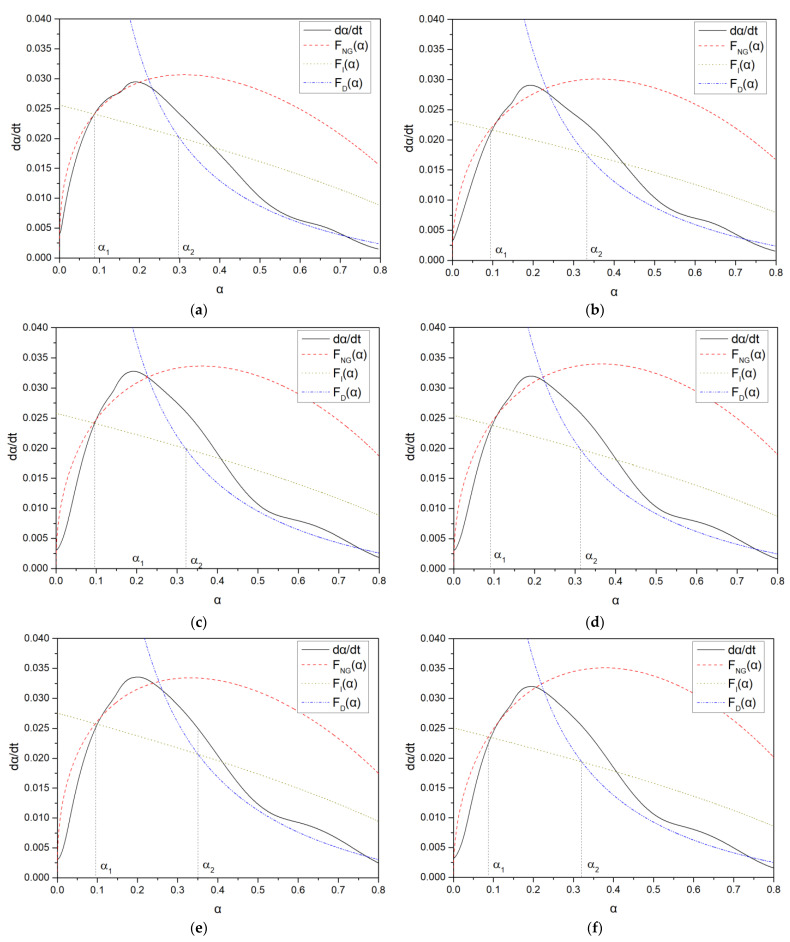
The curve of hydration degree α versus hydration rate dα/dt. (**a**) Blank (Without PCE); (**b**) PCE-1; (**c**) PCE-2; (**d**) PCE-3; (**e**) PCE-4; (**f**) PCE-5; (**g**) PCE-6; (**h**) PCE-7.

**Figure 13 polymers-15-02496-f013:**
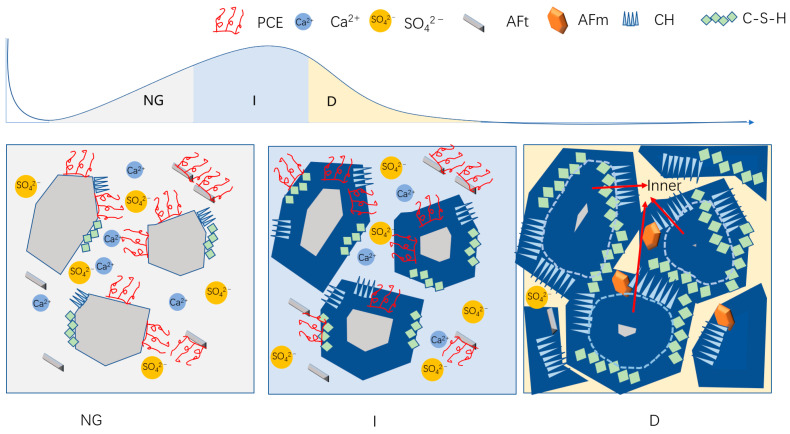
Schematic diagram of hydration.

**Table 1 polymers-15-02496-t001:** Chemical composition analysis of cement, /%.

CaO	SiO_2_	Al_2_O_3_	Fe_2_O_3_	MgO	SO_3_	Na_2_O	K_2_O	MnO	TiO_2_	Loss
63.79	19.80	5.12	3.65	2.30	2.49	0.30	0.31	0.12	0.16	1.85

**Table 2 polymers-15-02496-t002:** The mineral content of cement, /%.

C_3_S	C_2_S	C_3_A	C_4_AF
58.93	16.42	7.81	10.88

**Table 3 polymers-15-02496-t003:** Particle size distribution of cement.

X50 (μm)	<3 μm (%)	3~32 μm (%)	32~65 μm (%)	>65 μm (%)	>80 μm (%)
14.581	18.903	67.724	13.229	0.144	0.000

X50 is the particle size distribution where 50% of the particles have a diameter smaller than this value, meaning that the volume fraction of particles with diameters smaller than X50 accounts for 50% of the total particle volume.

**Table 4 polymers-15-02496-t004:** The molecular structure parameters of the PCE.

Samples	n(AA): n(TPEG)	SHP/%	Mn/Da	Mw/Da	Conversion/%	PDI	Rh/nm	Mark–Houwink α	ρ_s_	L_m_
PCE-1	1.5	1.5	22,461	36,142	68.3	1.61	4.62	0.26	0.31	14.41
PCE-2	3.0	1.5	24,335	42,692	87.1	1.75	5.00	0.16	0.23	16.32
PCE-3	4.5	1.5	25,247	44,843	89.6	1.78	5.10	0.17	0.17	16.46
PCE-4	6	1.5	26,263	47,852	90.3	1.82	5.46	0.21	0.13	16.90
PCE-5	4.5	0.5	48,328	92,690	90.0	1.92	6.94	0.35	0.16	34.03
PCE-6	4.5	1.0	35,489	60,722	90.9	1.71	5.85	0.33	0.17	22.29
PCE-7	4.5	2.0	19,123	30,452	90.0	1.59	4.37	0.30	0.19	11.18

**Table 5 polymers-15-02496-t005:** Rheological property data.

Sample	Solid Content/%	Flowability/mm	Plastic Viscosity/mPa·s	Yield Stress/MPa	Thixotropy Area/Pa·s·cm³
PCE-1	0.2	149	0.9862	14.2700	312.46
PCE-2	0.2	189	0.8232	7.6964	358.67
PCE-3	0.2	216	0.7303	3.9429	268.65
PCE-4	0.2	197	0.7425	7.0267	286.8
PCE-5	0.2	177	0.7442	8.9846	304.65
PCE-6	0.2	209	0.7752	3.3746	278.45
PCE-7	0.2	195	0.7568	7.2311	315.44

**Table 6 polymers-15-02496-t006:** Effect of different structure on fitting parameters of isothermal adsorption equation.

Sample	Langmuir			Freundlich			Temkin			R-P	
	q_e_/mg·g^−1^	K_L_/L·mg^−1^	R^2^	K_F_/(mg·g^−1^)·(L·mg^−1^)^1/n^	n_F_	R^2^	k_T_/ L·mol^−1^	b_T_/g·mol·mg^−1^·J^−1^	R^2^	k_RP_/ L·g^−1^	R^2^
PCE-1	0.8295	1.8620	0.9911	0.5202	3.4413	0.9714	27.8690	15,494.73	0.9910	3.7907	0.9901
PCE-2	1.4175	0.8915	0.9908	0.6622	2.4951	0.9858	8.7044	7897.65	0.9635	2.5646	0.9903
PCE-3	2.5466	0.6221	0.9790	0.9758	2.1070	0.9702	5.5048	4225.72	0.9707	1.5732	0.9910
PCE-4	3.3376	0.4169	0.9594	0.9823	1.7645	0.9638	3.5942	3183.23	0.9832	1.3805	0.9592
PCE-5	1.3109	2.2301	0.9993	0.8833	4.2279	0.9890	41.6606	10,423.41	0.9944	2.7751	0.9983
PCE-6	1.4486	1.7891	0.9935	0.9062	3.6167	0.9973	25.3350	8750.36	0.9964	4.8559	0.9964
PCE-7	1.3966	1.1434	0.9878	0.7340	2.8581	0.9728	10.8635	7980.01	0.9838	1.3491	0.9578

**Table 7 polymers-15-02496-t007:** Adsorption parameters of PCE on cement particle surface.

PCE	T/K	K_L_/L·mg^−1^	R_L_	∆G/kJ·mol^−1^	∆H/kJ·mol^−1^	∆S/J·mol^−1^·K^−1^
PCE-1	298	0.7552	0.3463	−24.14	−18.11	20.46
	308	0.6536	0.3797	−24.58		
	318	0.6317	0.3877	−25.29		
	328	0.4002	0.4999	−24.84		
PCE-2	298	0.1975	0.5584	−21.93	−14.31	17.08
	308	0.1682	0.6687	−21.47		
	318	0.1575	0.7175	−21.56		
	328	0.1411	0.7392	−21.93		
PCE-3	298	0.4266	0.4839	−23.00	−19.30	12.69
	308	0.3577	0.5279	−23.32		
	318	0.3461	0.5361	−23.99		
	328	0.2121	0.6535	−23.41		
PCE-4	298	0.8287	0.3463	−24.76	−24.70	0.69
	308	0.7061	0.3797	−25.18		
	318	0.4656	0.3877	−24.89		
	328	0.3447	0.4999	−24.86		
PCE-5	298	0.9363	0.2993	−26.57	−11.99	48.92
	308	2.4663	0.1396	−26.46		
	318	0.6908	0.3667	−27.55		
	328	8.9483	0.0428	−26.78		
PCE-6	298	2.9798	0.1183	−28.68	−15.70	43.57
	308	2.4454	0.1406	−29.13		
	318	2.0337	0.1644	−29.59		
	328	1.6645	0.1938	−29.97		
PCE-7	298	3.7597	0.0962	−29.32	−37.62	25.50
	308	1.1893	0.2517	−29.64		
	318	0.5980	0.4008	−29.78		
	328	0.8160	0.3290	−29.11		

**Table 8 polymers-15-02496-t008:** Fitting parameters of adsorption kinetics.

Sample	PFO				PSO				Elovich			ID		
	k_1_/ min^−1^	q_e1_/ mg·g^−1^	h_0_/ mg·(g·min)^−1^	R^2^	q_e2_/ mg·g^−1^	k_2_/ min^−1^	h_0_/ mg^2^·(g^2^·min)^−1^	R^2^	a/ mg·g^−1^·min^−1^	b/ g·mg^−1^	R^2^	k_ID_/ mg·g^−1^·min^−0.5^	C_ID_/ mg·g^−1^	R^2^
PCE-1	0.944	0.0629	0.0594	0.9311	1.0462	0.0896	0.0980	0.9568	0.8618	6.7907	0.9708	0.0715	0.2163	0.8090
PCE-2	1.617	0.0592	0.0957	0.9892	1.7749	0.0543	0.1709	0.9934	2.2001	4.3047	0.9876	0.1177	0.4001	0.7736
PCE-3	1.721	0.0546	0.0940	0.9799	1.8991	0.0460	0.1660	0.9947	1.5468	3.7758	0.9953	0.1268	0.4024	0.8045
PCE-4	1.820	0.0418	0.0760	0.9863	2.0849	0.0270	0.1175	0.9958	0.4146	2.7034	0.9925	0.1394	0.3300	0.8712
PCE-5	1.545	0.0343	0.0531	0.9381	1.7997	0.0249	0.0806	0.9658	0.2147	2.8412	0.9779	0.1207	0.2256	0.9145
PCE-6	1.634	0.0331	0.0541	0.9241	1.9148	0.0221	0.0810	0.9554	0.2020	2.6038	0.9692	0.1283	0.2248	0.9179
PCE-7	1.747	0.0682	0.1191	0.9874	1.8936	0.0629	0.2254	0.9973	5.7160	4.5343	0.9980	0.1259	0.4609	0.7607

**Table 9 polymers-15-02496-t009:** Heat evolution curve parameters of PCE with different carboxyl densities and main chain polymerization degrees.

Sample	t_0_/h	q_0_/ mW·g^−1^	Q_0_/J·g^−1^	t_1_/h	t_2_/h	K_2_/ mW·g^−1^·h^−1^	q_2_/ mW·g^−1^	t_3_/h	t_4_/h	q_3_/ mW·g^−1^	Q_3_/J·g^−1^	Q_0–3_/J·g^−1^
Blank	1.42	0.36	2.14	6.93	6.01	0.36	1.49	12.12	14.62	2.73	76.86	74.72
PCE-1	2.64	0.30	3.24	10.12	11.52	0.29	1.55	16.60	19.31	2.60	85.87	82.63
PCE-2	3.82	0.26	4.15	10.41	12.95	0.36	1.52	17.77	18.08	2.69	92.26	88.11
PCE-3	4.12	0.25	4.46	10.85	13.49	0.38	1.56	18.09	18.54	2.76	93.42	88.96
PCE-4	4.14	0.23	4.15	10.09	12.75	0.34	1.41	17.82	18.58	2.52	89.37	85.21
PCE-5	3.73	0.26	4.03	10.44	13.09	0.35	1.47	17.99	18.56	2.62	89.91	85.88
PCE-6	4.09	0.25	4.27	10.78	4.09	0.37	1.50	18.08	18.15	2.63	90.06	85.80
PCE-7	3.92	0.26	4.21	10.73	14.21	0.36	1.70	18.44	19.10	2.78	95.85	91.65

Note: t_0_, t_1_, t_2_, t_3_, and t_4_ are the times for the end of the induction period or the start of the NG process, the start of the I process, the time corresponding to the maximum slope of acceleration period curve, the end of acceleration period, and the start of D process, respectively; q_0_, q_2_, q_3_, Q_0_, Q_3_, and Q_0–3_ are the heat release rate and cumulative heat release of the corresponding points; and K_2_ is the transversal slope between 0 and 2.

**Table 10 polymers-15-02496-t010:** Hydration kinetic parameters of PCE with different carboxyl group densities and main chain polymerization degrees.

Sample	Q_max_/J·g^−1^	t_50_/h	NG		I	D	α_1_	α_2_	α_3_	Δα_1_	Δα_2_
			K_NG_/h^−1^	n	K_I_/μm·h^−1^	K_D_/μm^2^·h^−1^					
Blank	333.33	24.73	0.0404	1.59	0.0085	0.0019	0.01	0.09	0.30	0.08	0.29
PCE-1	322.58	25.26	0.0374	1.80	0.0077	0.0019	0.01	0.09	0.33	0.08	0.32
PCE-2	303.03	23.36	0.0419	1.82	0.0085	0.0020	0.01	0.09	0.31	0.08	0.30
PCE-3	303.03	23.39	0.0425	1.85	0.0086	0.0021	0.01	0.09	0.32	0.08	0.31
PCE-4	270.27	19.89	0.0431	1.67	0.0092	0.0025	0.02	0.09	0.35	0.08	0.34
PCE-5	294.12	22.62	0.0423	1.90	0.0084	0.0020	0.01	0.09	0.32	0.07	0.31
PCE-6	294.12	25.29	0.0429	1.86	0.0085	0.0020	0.02	0.09	0.31	0.07	0.29
PCE-7	312.50	24.28	0.0421	1.94	0.0082	0.0020	0.01	0.09	0.32	0.07	0.31

**Table 11 polymers-15-02496-t011:** Degree of hydration and characteristics of cement hydration products.

Sample	Hydration Time 2 h		Hydration Time 12 h		Hydration Time 1 d		Hydration Time 3 d	
	Hydration Degree	Morphology	Hydration Degree	Morphology	Hydration Degree	Morphology	Hydration Degree	Morphology
Blank	0.0101	Corrosion pit	0.2272	Subtle hydration product, no needle-like	0.4650	Needle-like, length 1~2 μm	0.7433	Gel substrate
PCE-1	0.0072	Corrosion pit	0.1383		0.4230	Fuzzy, non-needle-like	0.7440	substrate, needle-like
PCE-2	0.0064	Corrosion pit	0.1227		0.4336	Needle-like, length 1 μm	0.7653	substrate, needle-like
PCE-3	0.0065	Corrosion pit	0.1319		0.4327	Needle-like, few, length 1 μm	0.7591	substrate, needle-like
PCE-4	0.0064	Corrosion pit	0.1488		0.4571	Needle-like, few, length 1 μm	0.7957	Gel substrate, rod-like, needle-like
PCE-5	0.0067	Corrosion pit	0.1267		0.4306	Needle-like, few, length 1 μm	0.7596	Gel substrate, rod-like, needle-like
PCE-6	0.0064	Corrosion pit	0.1246		0.4256	Needle-like, length 1 μm	0.7463	Gel substrate, rod-like, needle-like
PCE-7	0.0062	Corrosion pit	0.1159		0.4262	Needle-like, length 2 μm	0.7532	Gel substrate, rod-like, needle-like

**Table 12 polymers-15-02496-t012:** SEM of hydration products of samples with different carboxyl group densities and main chain polymerization degrees (hydration time: 2 h and 12 h).

Sample	Hydration Time 2 h	Hydration Time 12 h
Blank	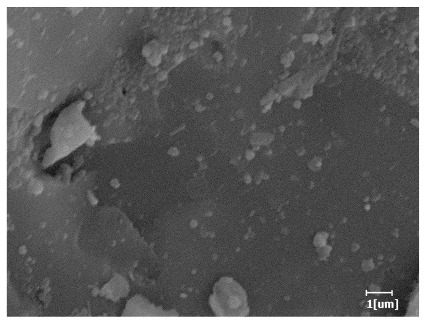	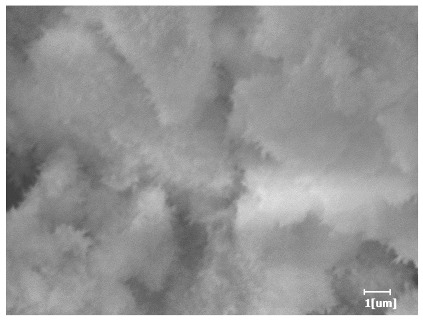
PCE-1	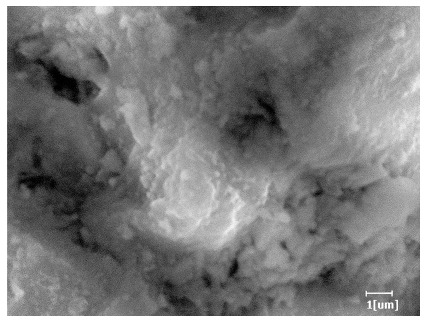	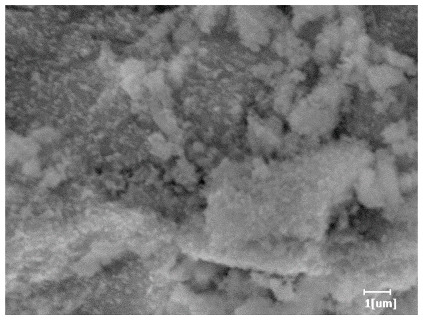
PCE-2	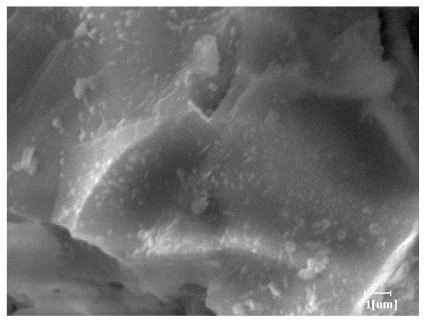	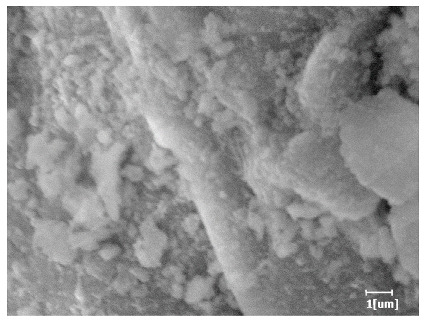
PCE-3	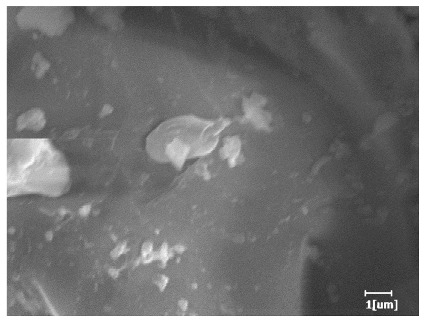	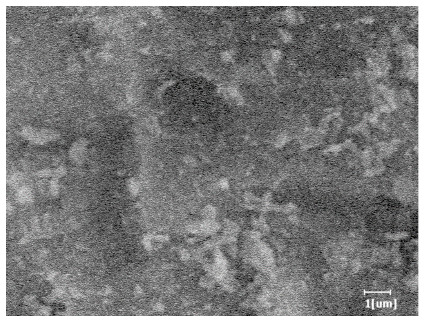
PCE-4	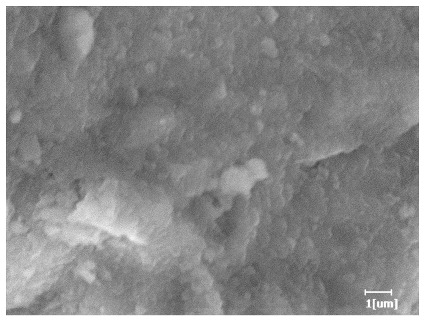	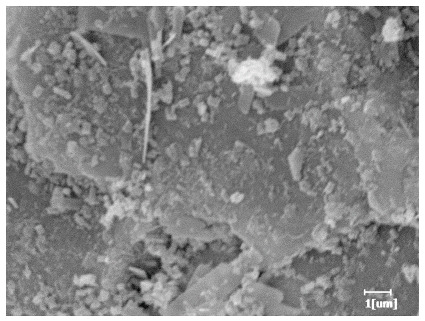
PCE-5	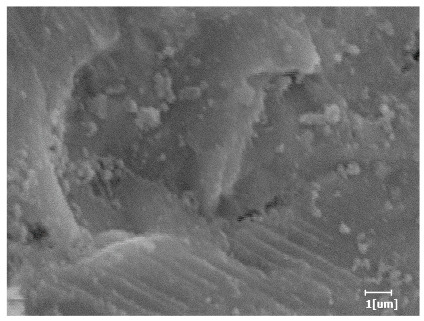	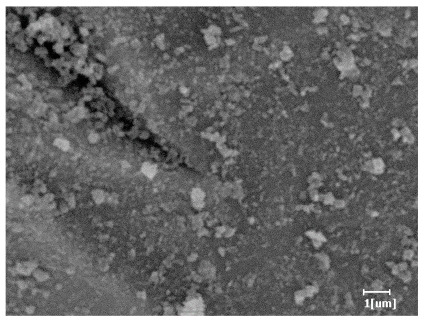
PCE-6	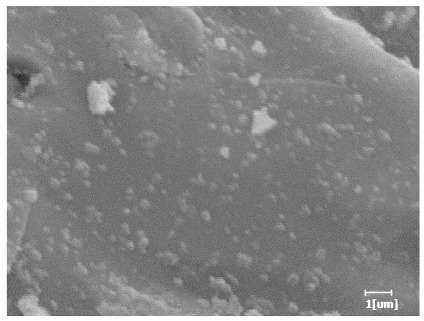	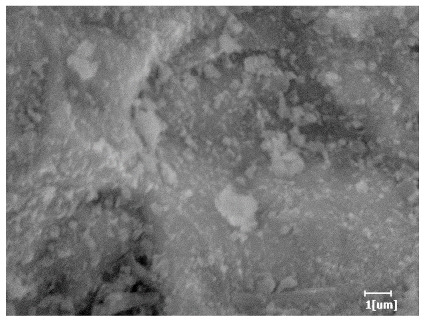
PCE-7	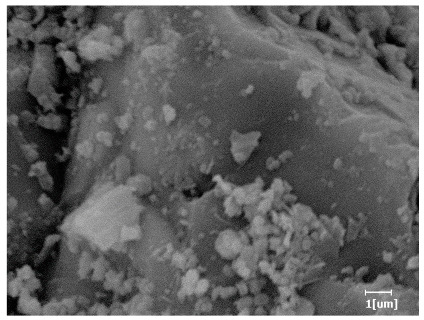	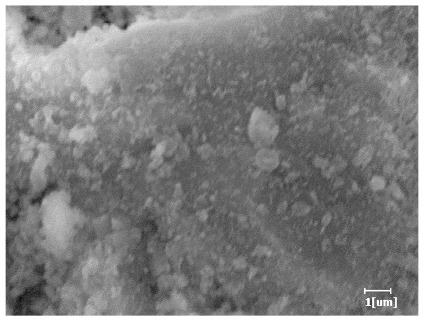

**Table 13 polymers-15-02496-t013:** SEM of hydration products of samples with different carboxyl group densities and main chain polymerization degrees (hydration time: 1 day and 3 days).

Sample	Hydration Time 1 d	Hydration Time 3 d
Blank	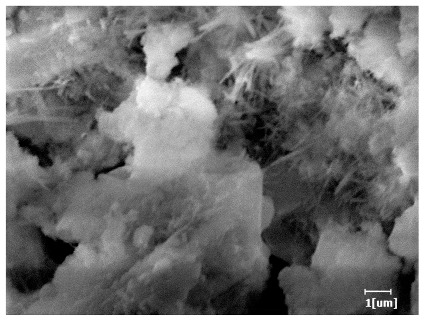	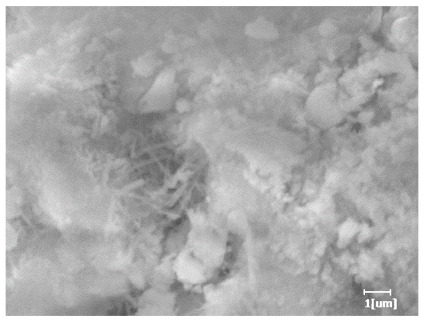
PCE-1	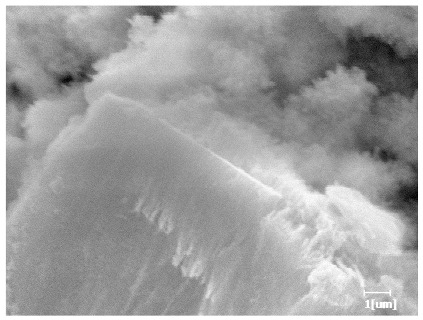	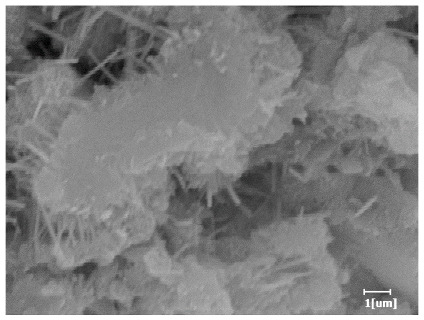
PCE-2	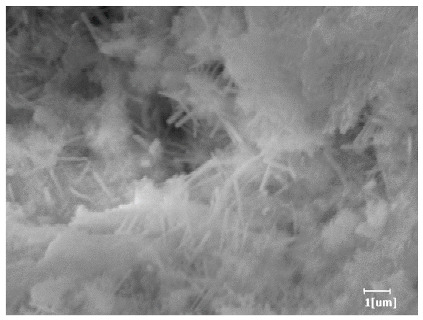	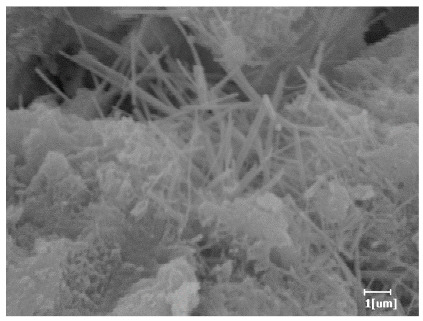
PCE-3	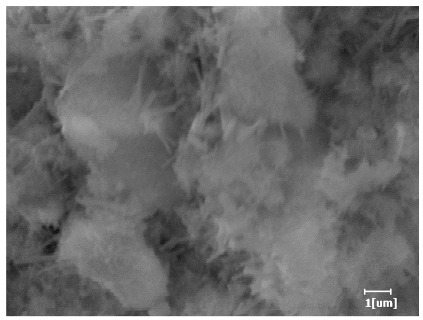	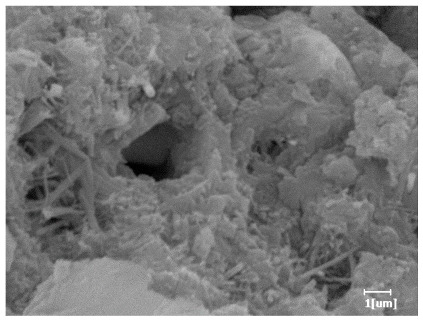
PCE-4	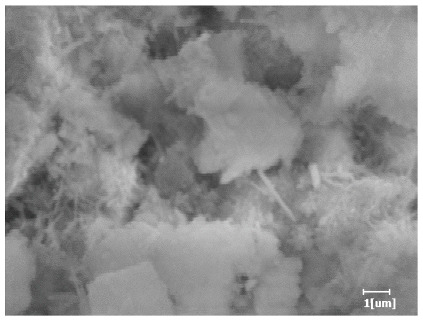	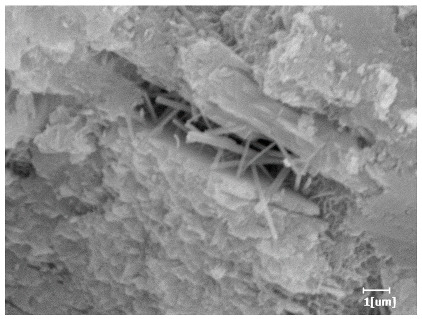
PCE-5	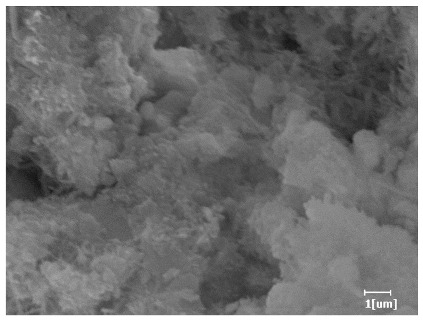	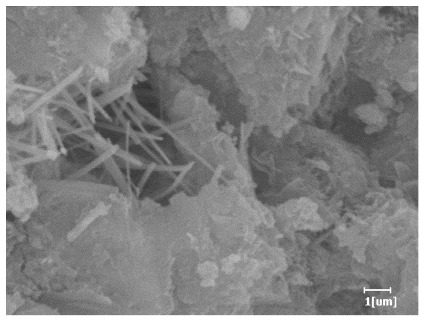
PCE-6	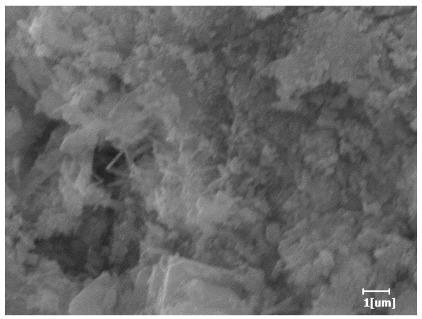	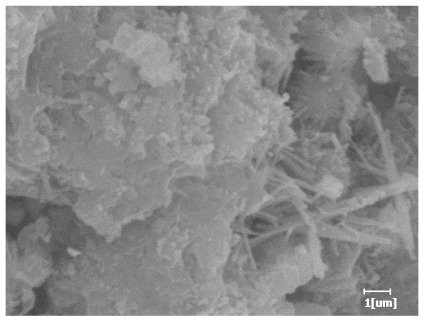
PCE-7	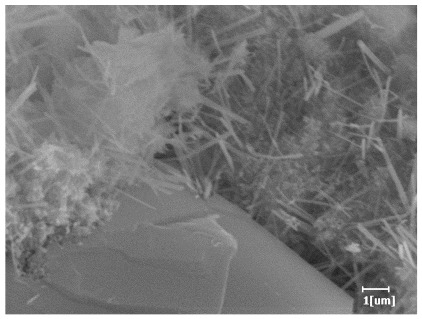	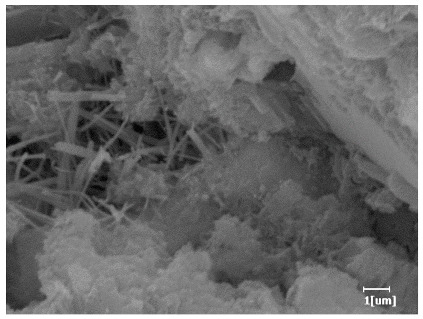

## Data Availability

Data will be made available on request.

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
