# Peer review of "Study on the Effect of Polycarboxylate Ether Molecular Structure on Slurry Dispersion, Adsorption, and Microstructure"

_polymers, 2023, doi:10.3390/polym15112496_

Round 1

Reviewer 1 Report

Subject:  Corrections and comments made to paper entitled: “Study on the effect of polycarboxylate ether molecular structure on slurry dispersion, adsorption, and microstructure” (Manuscript ID polymers-2408495).

Dear Madam/Sir,

Please find below the corrections and comments made to the paper above.

For a better understanding of the readers, provide techniques and conditions utilized to obtain the results shown in Tables 1, 2 and 3.

For the rheology measurements, add the geometry utilized for the study.

A concern:

The design and proportions of the specimens normally remain the same for results comparisons. Nevertheless, heat of hydration test was conducted at a different water to cement ration than the rest of the study.

1.       Specify if this is the only change in this parameter or if other studies vary some parameters of the paste specimen fabrications.

2.       Why the relationship varies from the other analysis.

In page 9, units mush has a correct format. Review lines 339 and 341.

The result data (dots or lines) is overlapping in the figures, so the information can not be review. Enlarge the graph or divide it for a better understanding. See figures 7, 9, 11 and 12.

Figure 10 is very general and it can be claim by other authors due to similarities. Add references or more details in the graph to be more unique for the authors.

Regards!!

Author Response

Dear Sir/Madam,

Thank you very much for taking the time to review our manuscript amidst your busy schedule. We greatly appreciate your valuable feedback. In response to the specific points you raised, we have addressed them as follows:

Q1: For a better understanding of the readers, provide techniques and conditions utilized to obtain the results shown in Tables 1, 2 and 3.

A1: The experimental methods and equipment models used for obtaining the data results in Tables 1, 2, and 3 are provided below. These details have been incorporated into the latest version of the manuscript: “The chemical composition analysis of cement was determined according to the Chinese standard GB/176-2017 "Methods for chemical analysis of cement." The mineral composition of the cement was determined based on the data results in Table 1 according to the Chinese standard GB/T 21372-2008 "Portland cement clinker." The particle size distribution of the cement was measured using the laser particle size analyzer from Jinan Winner Particle Instrument Stock Co., Ltd., with the model name Winner 3000.”

Q2: For the rheology measurements, add the geometry utilized for the study.

A2: We utilized the Rheolab QC instrument, manufactured by Anton Paar in Austria. The hollow cylinder formed by the rotor and the measuring cup in this instrument represents the laminar flow region for rheological testing. We extracted the diameter of the rotor and the inner diameter of the measuring cup in order to calculate the corresponding height of the laminar flow. “The rotor diameter is 39mm, the cylinder diameter is 42mm, and the height of the rheological laminar flow is 1.5mm.”

Q3-1: The design and proportions of the specimens normally remain the same for results comparisons. Nevertheless, heat of hydration test was conducted at a different water to cement ration than the rest of the study.

1.Specify if this is the only change in this parameter or if other studies vary some parameters of the paste specimen fabrications.

A3-1: We apologize for the writing mistake. Our hydration heat test was also conducted with a water-to-cement ratio of 0.29. This information has been updated in line 283 of the manuscript.

Q3-2: Why the relationship varies from the other analysis.

A3-2: The hydration heat variation is influenced by the quantity and concentration of the pore solution, and it exhibits different changes in different stages. Unlike the flowability test, the hydration stages include the induction period and the acceleration period, each with distinct variations. Particularly during the acceleration period, the consumption of the saturated solution due to the volume and crystallization nucleation and growth differs, resulting in different outcomes. This is explained in lines 601 to 604 of the manuscript.

Q4: In page 9, units mush has a correct format. Review lines 339 and 341.

A4: The unit format has been corrected and assigned superscripts.

Q5: The result data (dots or lines) is overlapping in the figures, so the information can not be review. Enlarge the graph or divide it for a better understanding. See figures 7, 9, 11 and 12.

A5: Thank you for pointing out the drawback of dense curves in the figures that may hinder readability. As a result, we have redesigned the graphs accordingly. Figures 7 and 9 have been divided for easier viewing, while Figure 11 has been enlarged with a local zoom-in feature to display each curve clearly. Additionally, Figure 12 has been scaled up in size. For more specific details, please refer to the revised manuscript.

Q6: Figure 10 is very general and it can be claim by other authors due to similarities. Add references or more details in the graph to be more unique for the authors.

A6:  Based on your suggestion, we have added the references.

The above is our detailed response to your questions. We sincerely appreciate your valuable feedback on our paper. We kindly request you to review and consider our replies. Thank you once again.

Best wishes.

Reviewer 2 Report

The article is devoted to an important problem: ensuring the possibility of controlling and regulating the processes of hydration and the rheological properties of Portland cement in the presence of a plasticizer due to its directed synthesis.

The article provides a good literature review on this issue; a large amount of research has been carried out; interesting results have been obtained.

However, in order to improve the presentability of the manuscript, it is recommended to take into account the following comments:

Line 217. Table3. Please, explain, what does means "X50" when describing the particle size distribution of cement;

Line 255. Please, give a transcript of the abbreviation "TOC";

Lines 291–292. The designation of parameters of "ρ_s" and "L_m" differ in the text and in formulas (7) and (8) - "ρs" and "Lm". Please use the same notations for these parameters;

Formulas 9–10. What is "??", "?0" and "?0"?

Line 563. In Figure 10, the x-axis is recommended to show specific time intervals;

Line 652. Table 12. Text is illegible on SEM micrographs, scale bar is almost invisible;

Line 706. Description of AFm phase and its decoding is missing in the text. It is recommended to add information about AFm.

Author Response

Dear Sir/Madam,

Thank you very much for taking the time to review our manuscript amidst your busy schedule. We greatly appreciate your valuable feedback. In response to the specific points you raised, we have addressed them as follows:

Q1: Line 217. Table3. Please, explain, what does means "X50" when describing the particle size distribution of cement.

A1: X50 is the particle size distribution where 50% of the particles have a diameter smaller than this value, meaning that the volume fraction of particles with diameters smaller than X50 accounts for 50% of the total particle volume. This explanation has been updated in the latest version of the manuscript. 

Q2: Line 255. Please, give a transcript of the abbreviation "TOC".

A2: Thank you for your reminder. TOC stands for Total Organic Carbon test. This explanation has been updated in the latest version of the manuscript.

Q3: Lines 291–292. The designation of parameters of "ρ_s" and "L_m" differ in the text and in formulas (7) and (8) - "ρs" and "Lm". Please use the same notations for these parameters.

A3: There was an output error in the formatting of two symbols. We have rectified the symbols of these two parameters to match those in equations (7) and (8).

Q4: Formulas 9–10. What is "??", "?0" and "?0"?

A4: Thank you for the reminder. Formula 9 has been updated to RL = 1/(1 + KLC0). Moreover, in formulas 10 and 11, K0 has been changed to KL, and explanations for C0 and RL have been provided. C0 (mg·L-1)represents the initial concentration of the adsorbate in the solution ,  while RL (dimensionless) stands for the separation factor.

Q5: Line 563. In Figure 10, the x-axis is recommended to show specific time intervals.

A5: Thank you for your reminder. We have added specific time intervals to the x-axis of Figure 10.

Q6: Line 652. Table 12. Text is illegible on SEM micrographs, scale bar is almost invisible.

A6: We have divided Table 12 into separate sections based on hydration time, specifically 1 hour and 12 hours, as well as 1 day and 3 days. Additionally, we have enlarged the SEM images so that the text and scale can be viewed clearly.

Q7: Line 706. Description of AFm phase and its decoding is missing in the text. It is recommended to add information about AFm.

A7: Thank you for your suggestion. AFm refers to monosulphate phase, which undergoes a transformation from AFt to AFm in the absence of sulfate ions. The shape also changes from a fibrous structure to a hexagonal plate-like morphology. We have provided additional explanations in lines 718-720 to supplement this information.

The above is our detailed response to your questions. We sincerely appreciate your valuable feedback on our paper. We kindly request you to review and consider our replies. Thank you once again.

Best wishes.